# LARGE LANGUAGE MODELS ARE HUMAN-LEVEL PROMPT ENGINEERS

**Yongchao Zhou**[1,2,*], **Andrei Ioan Muresanu**[2,3,*], **Ziwen Han**[1,2,*], **Keiran Paster**[1,2],
**Silviu Pitis**[1,2], **Harris Chan**[1,2], **Jimmy Ba**[1,2]
[1]University of Toronto  [2]Vector Institute  [3]University of Waterloo  [*]Equal contribution
{yczhou,hanziwen,keirp,spitis,hchan,jba}@cs.toronto.edu
{andrei.muresanu}@uwaterloo.ca

## ABSTRACT

By conditioning on natural language instructions, large language models (LLMs) have displayed impressive capabilities as general-purpose computers. However, task performance depends significantly on the quality of the prompt used to steer the model, and most effective prompts have been handcrafted by humans. Inspired by classical program synthesis and the human approach to prompt engineering, we propose *Automatic Prompt Engineer*[1] (APE) for automatic instruction generation and selection. In our method, we treat the instruction as the "program," optimized by searching over a pool of instruction candidates proposed by an LLM in order to maximize a chosen score function. To evaluate the quality of the selected instruction, we evaluate the zero-shot performance of another LLM following the selected instruction. Extensive experiments show that our automatically generated instructions outperform the prior LLM baseline by a large margin and achieve better or comparable performance to the instructions generated by human annotators on 24/24 Instruction Induction tasks and 17/21 curated BIG-Bench tasks. We conduct extensive qualitative and quantitative analyses to explore the performance of APE. We show that APE-engineered prompts are able to improve few-shot learning performance (by simply prepending them to standard in-context learning prompts), find better zero-shot chain-of-thought prompts, as well as steer models toward truthfulness and/or informativeness. [2]

## 1 INTRODUCTION

The combination of scale and attention-based architectures has resulted in language models possessing an unprecedented level of generality (Kaplan et al., 2020; Vaswani et al., 2017). These so-called "large language models" (LLMs) have shown remarkable, often superhuman, capabilities across a diverse range of tasks, including both zero-shot and few-shot setups (Brown et al., 2020; Srivastava et al., 2022). With generality, however, there comes a question of control: how can we make LLMs do what we want them to do?

To answer this question and steer LLMs toward desired behaviors, recent work has considered fine-tuning (Ouyang et al., 2022; Ziegler et al., 2019), in-context learning (Brown et al., 2020), and several forms of prompt generation (Gao, 2021), including both differentiable tuning of soft prompts (Qin & Eisner, 2021; Lester et al., 2021) and natural language prompt engineering (Reynolds & McDonell, 2021). The latter is of particular interest, as it provides a natural interface for humans to communicate with machines and may be of great relevance not only to LLMs but to other generalist models such as prompted image synthesizers (Rombach et al., 2022; Ramesh et al., 2022), for which public interest in prompt design and generation has also emerged (see Appendix A for examples).

Behind this interest is the fact that plain language prompts do not always produce the desired results, even when those results are possible to produce with alternative instructions. Thus, human users must experiment with a wide range of prompts to elicit desired behaviors, as they have little knowledge of how compatible instructions are with a particular model. We can understand this by viewing LLMs as black-box computers that execute programs specified by natural language instructions: while they

---

[1]We define "prompt engineering" as optimizing the language in a prompt in order to elicit the best possible performance. Notably, this does not include prompts that chain multiple LLM queries together or give the LLM access to external tools.

[2] Our code is available at https://github.com/keirp/automatic_prompt_engineer.

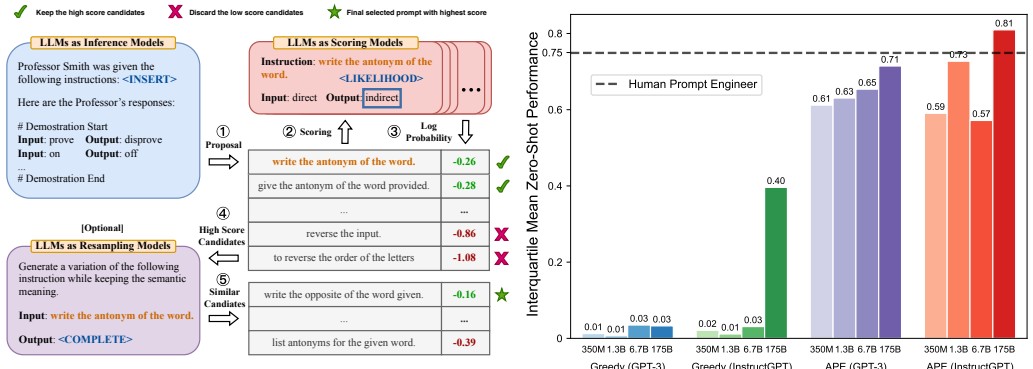

(a) Automatic Prompt Engineer (APE) workflow      (b) Interquartile mean across 24 tasks

Figure 1: (a) Our method, **Automatic Prompt Engineer (APE)**, automatically generates instructions for a task that is specified via output demonstrations: it generates several instruction candidates, either via direct inference or a recursive process based on semantic similarity, executes them using the target model, and selects the most appropriate instruction based on computed evaluation scores. (b) As measured by the interquartile mean across the 24 NLP tasks introduced by Honovich et al. (2022), APE is able to surpass human performance when using the InstructGPT model (Ouyang et al., 2022).

can execute a broad range of natural language programs, the way these programs are processed may not be intuitive for humans, and the quality of instruction can only be measured when executing these instructions on a downstream task (Sanh et al., 2022; Wei et al., 2021).

To reduce the human effort involved in creating and validating effective instructions, we propose a novel algorithm using LLMs to generate and select instructions automatically. We call this problem *natural language program synthesis* and propose to address it as a black-box optimization problem using LLMs to generate and search over heuristically viable candidate solutions. In doing so, we leverage the generalist capabilities of LLMs in three ways. First, we use an LLM as an inference model (Ellis et al., 2021; Honovich et al., 2022) to generate instruction candidates based on a small set of demonstrations in the form of input-output pairs. Next, we guide the search process by computing a score for each instruction under the LLM we seek to control. Finally, we propose an iterative Monte Carlo search method where LLMs improve the best candidates by proposing semantically similar instruction variants. Intuitively, our algorithm asks LLMs to generate a set of instruction candidates based on demonstrations and then asks them to assess which instructions are more promising. We call our algorithm Automatic Prompt Engineer (APE). **Our main contributions are:**

- We frame instruction generation as natural language program synthesis, formulate it as a black-box optimization problem guided by LLMs, and propose both a naive and an iterative Monte Carlo search methods to approximate the solution.

- Our proposed method, APE, achieves human-level performance on zero-shot learning with model-generated instructions on 24/24 Instruction Induction and 17/21 Big-Bench tasks.

- We provide extensive qualitative and quantitative analyses exploring various facets of APE, and demonstrate applications of APE for improving few-shot learning, finding better zero-shot chain of thought prompts, and steering LLMs toward desired behaviors such as truthfulness and/or informativeness.

## 2 RELATED WORK

**Large Language Models**    Scaling up transformer-based language models in terms of model size, training data, and training compute has been shown to predictably improve performance on a wide range of downstream NLP tasks (Vaswani et al., 2017; Devlin et al., 2018; Brown et al., 2020). Many emergent abilities (Wei et al., 2022a) of LLMs have been discovered as a result of this scaling, including few-shot in-context learning, zero-shot problem solving, chain of thought reasoning, instruction following, and instruction induction (Cobbe et al., 2021; Wei et al., 2022b; Kojima et al.,

2022; Sanh et al., 2022; Wei et al., 2021; Ouyang et al., 2022; Honovich et al., 2022). In this paper, we view LLMs as black-box computers that execute programs specified by natural language instructions and investigate how to control an LLM's behavior using model-generated instructions.

**Prompt Engineering**  Prompting offers a natural and intuitive interface for humans to interact with and use generalist models such as LLMs. Due to its flexibility, prompting has been widely used as a generic method for NLP tasks (Schick & Schütze, 2021; Brown et al., 2020; Sanh et al., 2022). However, LLMs require careful prompt engineering, either manually (Reynolds & McDonell, 2021) or automatically (Gao et al., 2021; Shin et al., 2020), as models do not seem to understand the prompts in the same way a human would (Webson & Pavlick, 2021; Lu et al., 2021). Though many successful prompt tuning methods perform optimization over a continuous space using gradient-based methods (Liu et al., 2021; Qin & Eisner, 2021; Lester et al., 2021), this becomes less practical with scale, as computing gradients becomes increasingly expensive and access to models shifts to APIs that may not provide gradient access. In our paper, we borrow components from discrete prompt search methods, such as prompt generation (Gao et al., 2021; Ben-David et al., 2021), prompt scoring (Davison et al., 2019) and prompt paraphrasing (Jiang et al., 2020; Yuan et al., 2021) to optimize instructions by searching directly in the natural language hypothesis space. As compared to this past work, which uses specialized models for each component and leans heavily on human templates, we show that the entire search can be conducted by a single LLM.

**Program Synthesis**  Program synthesis involves the automatic search over a "program space" to find a program satisfying a particular specification (Gulwani et al., 2017). Modern program synthesis admits a wide variety of specifications, including input-output examples (Ellis et al., 2021; Wong et al., 2021) and natural language (Jain et al., 2022). The range of feasible program spaces to search over has also grown, from historically restrictive domain-specific languages to general-purpose programming languages (Austin et al., 2021). In contrast to prior approaches that require a suitable structured hypothesis space and library of components (Liang et al., 2010; Ellis et al., 2018), we leverage the structure provided by LLMs to search over the space of natural language programs. Using inference models is a standard practice to speed up the search by restricting the search space to a limited space of possible expressions (Menon et al., 2013; Lee et al., 2018; Devlin et al., 2017; Ellis et al., 2021). Inspired by this, we use LLMs as approximate inference models to generate program candidates based on a small set of demonstrations. Unlike classical program synthesis, our inference models do not require any training and generalize well to various tasks.

## 3 NATURAL LANGUAGE PROGRAM SYNTHESIS USING LLMS

We consider a task specified by a dataset $\mathcal{D}_{\text{train}} = \{(Q, A)\}$ of input/output demonstrations sampled from population $\mathcal{X}$, and a prompted model $\mathcal{M}$. The goal of natural language program synthesis is to find a single instruction $\rho$ such that, when $\mathcal{M}$ is prompted with the concatenation $[\rho; Q]$ of instruction and a given input, $\mathcal{M}$ produces the corresponding output $A$. More formally, we frame this as an optimization problem, where we seek instruction $\rho$ that maximizes the expectation of some per-sample score $f(\rho, Q, A)$ over possible $(Q, A)$:

$$\rho^{\star} = \arg\max_{\rho} f(\rho) = \arg\max_{\rho} \mathbb{E}_{(Q,A)} \left[ f(\rho, Q, A) \right] \tag{1}$$

Note that in general, $Q$ may be the empty string, such that we are optimizing $\rho$ as a prompt that directly produces outputs $\{A\}$. While this task has been widely attempted by humans, we have little knowledge of how compatible any particular instruction is with model $\mathcal{M}$. Thus, we propose to treat this human-intractable question as a black-box optimization process guided by LLMs. Our algorithm, APE, uses LLMs in each of two key components, proposal and scoring. As shown in Figure 1 and summarized in Algorithm 1, APE first proposes a few candidate prompts, and then filters/refines the candidate set according to a chosen score function, ultimately choosing the instruction with the highest score. We discuss options for proposal and scoring next.

### 3.1 INITIAL PROPOSAL DISTRIBUTIONS

Due to the infinitely large search space, finding the right instruction can be extremely difficult, which has rendered natural language program synthesis historically intractable. Recent progress in NLP has shown language models are very good at generating diverse natural language text. Therefore, we

---

**Algorithm 1** Automatic Prompt Engineer (APE)

---

**Require:** $\mathcal{D}_{\text{train}} \leftarrow \{(Q, A)\}_n$: training examples, $f : \rho \times \mathcal{D} \mapsto \mathbb{R}$: score function
1: Use LLM to sample instruction proposals $\mathcal{U} \leftarrow \{\rho_1, ..., \rho_m\}$. (See Section 3.1)
2: **while** not converged **do**
3:     Choose a random training subset $\widetilde{\mathcal{D}}_{\text{train}} \subset \mathcal{D}_{\text{train}}$.
4:     **for all** $\rho$ in $\mathcal{U}$ **do**
5:         Evaluate score on the subset $\widetilde{s} \leftarrow f(\rho, \widetilde{\mathcal{D}}_{\text{train}})$ (See Section 3.2 )
6:     **end for**
7:     Filter the top k% of instructions with high scores $\mathcal{U}_k \subset \mathcal{U}$ using $\{\widetilde{s}_1, ..., \widetilde{s}_m\}$
8:     Update instructions $\mathcal{U} \leftarrow \mathcal{U}_k$ or use LLM to resample $\mathcal{U} \leftarrow \text{resample}(\mathcal{U}_k)$ (See Section 3.3)
9: **end while**
    **Return** instruction with the highest score $\rho^\star \leftarrow \arg\max_{\rho \in \mathcal{U}_k} f(\rho, \mathcal{D}_{\text{train}})$

---

consider leveraging a pretrained LLM to propose a good set $\mathcal{U}$ of candidate solutions that will guide our search procedure. While random samples from LLMs are unlikely to produce the desired $(Q, A)$ pairs, we can instead ask the LLM to approximately infer the most likely instructions with a high score, given the input/output demonstrations; i.e., to approximately sample from $P(\rho \mid \mathcal{D}_{\text{train}}, f(\rho)$ is high$)$.

**Forward Mode Generation** We consider two approaches to generate high-quality candidates from $P(\rho \mid \mathcal{D}_{\text{train}}, f(\rho)$ is high$)$. First, we adopt an approach based on "forward" mode generation by translating this distribution $P(\rho \mid \mathcal{D}_{\text{train}}, f(\rho)$ is high$)$ into words. For example, in our instruction induction experiments (Subsection 4.1), we follow Honovich et al. (2022) and prompt the LLM using Figure 2 (Top).

**Reverse Mode Generation** Although the "forward" model works out of the box for most of the pretrained LLMs, translating $P(\rho \mid \mathcal{D}_{\text{train}}, f(\rho)$ is high$)$ into words requires custom engineering across different tasks. This is because while instructions are typically found in the beginning of passages, the "forward" model only generates text from left to right, which requires the instruction to be predicted at the end of the prompt. Therefore, we desire a more flexible approach such that the instruction can be anywhere in the text. To address this, we consider "reverse" mode generation, which uses an LLM with infilling capabilities—e.g., T5 (Raffel et al., 2020), GLM (Du et al., 2022), and InsertGPT (Bavarian et al., 2022)—to infer the missing instructions. Our "reverse" model directly samples from $P(\rho \mid \mathcal{D}_{\text{train}}, f(\rho)$ is high$)$ by filling in the blank. We show an example of the such template in Figure 2 (Middle).

**Customized Prompts** Note that depending on the score function being used, there may exist more appropriate prompts than the samples above. For example, in our TruthfulQA experiments, we start with the human-designed instructions from the original dataset (Lin et al., 2022) and ask the the "reverse" model to propose initial instruction samples that fit the missing context (Figure 2 (Bottom)).

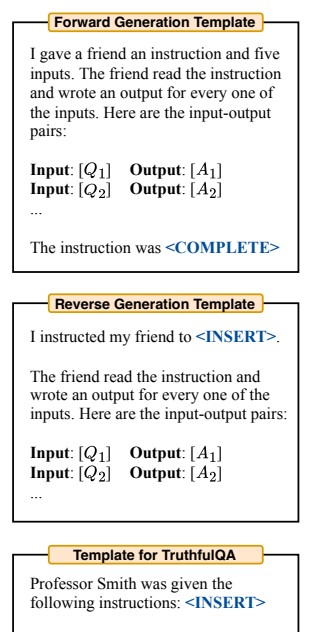

Figure 2: Prompts for LLMs

## 3.2 SCORE FUNCTIONS

To cast our problem as black-box optimization, we choose a score function that accurately measures the alignment between the dataset and the data the model generates. In our instruction induction experiments, we consider two potential score functions, described below. In the TruthfulQA experiments, we focused primarily on automated metrics proposed in Lin et al. (2022), similar to the execution accuracy. In each case, we evaluate the quality of a generated instruction using Equation (1), and take the expectation over a held-out test dataset $\mathcal{D}_{\text{test}}$.

**Execution accuracy** First, we consider evaluating the quality of an instruction $\rho$ using the execution accuracy metric proposed by Honovich et al. (2022), which we denote as $f_{\text{exec}}$. In most cases,

execution accuracy is simply defined as the 0-1 loss, $f(\rho, Q, A) = \mathbb{1}[\mathcal{M}([\rho; Q]) = A]$. On some tasks, execution accuracy takes into account invariants; e.g., it may be an order invariant set matching loss, as described in Appendix A of Honovich et al. (2022).

**Log probability** We further consider a softer probabilistic score function, which we hypothesize might improve optimization by providing a more fine-grained signal when searching over low-quality instruction candidates. In particular, we consider the log probability of the desired answer given the instruction and question under the target model $\mathcal{M}$, which on a per sample basis, is $\log P(A \mid [\rho; Q])$.

**Efficient score estimation** Estimating the score by computing the score over the entire training dataset for all instruction candidates can be expensive. To reduce the computation cost, we adopt a filtering scheme where a promising candidate receives more computation resources while a low-quality candidate receives less computation. It can be achieved by using a multi-stage computation strategy on lines 2-9 Algorithm 1. We first evaluate all candidates with a small subset of the training dataset. For the candidates with a score greater than a certain threshold, we sample and evaluate a new non-overlapping subset from the training dataset to update the moving average of the score. Then, we repeat this process until a small set of candidates is left, which are evaluated on the entire training dataset. This adaptive filtering scheme significantly improves the computation efficiency by keeping the exact computation costs for the high-quality samples and drastically reducing the computation costs for low-quality candidates. We note that a similar score estimation scheme has been used in previous works (Li et al., 2022; Maclaurin & Adams, 2015).

### 3.3 ITERATIVE PROPOSAL DISTRIBUTIONS

Despite our attempt to directly sample high-quality initial instruction candidates, it could be the case that the method described in Subsection 3.1 fails to produce a good proposal set $\mathcal{U}$, either because it lacks of diversity or does not contain any candidates with a suitably high score. In case of such challenges, we explore an iterative process for resampling $\mathcal{U}$.

**Iterative Monte Carlo Search** Instead of only sampling from the initial proposal, we consider exploring the search space locally around the current best candidates. This allows us to generate new instructions that are more likely to be successful. We call this variant *iterative APE*. At each stage, we evaluate a set of instructions and filter out candidates with low scores. Then, an LLM is asked to generate new instructions similar to those with high scores. We provide the prompt used for resampling in Figure 3. Figure 6 (Right) shows that although this approach improves the overall quality of the proposal set $\mathcal{U}$, the highest scoring instruction tends to remain the same with more stages. We conclude iterative generation provides marginal improvement over the relative simplicity and effectiveness of the generative process described in Subsection 3.1. Therefore, we use APE without iterative search as default unless otherwise stated.

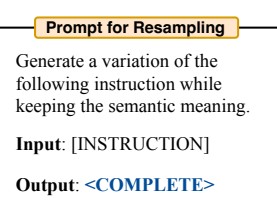

Figure 3: Resampling

## 4 LARGE LANGUAGE MODELS ARE HUMAN-LEVEL PROMPT ENGINEERS

This section examines how APE can guide LLMs to desired behaviors. We investigate from four perspectives: zero-shot performance, few-shot in-context learning performance, zero-shot chain-of-thought reasoning, and truthfulness. Our experiments show that APE can find prompts that improve task performance, performing equal to or even better than those authored by humans. APE also often produces insightful tricks for how to best prompt language models that can be successfully transferred to new tasks (see Section 4.3).

### 4.1 INSTRUCTION INDUCTION

We assess the effectiveness of zero-shot and few-shot in-context learning on 24 instruction induction tasks proposed in Honovich et al. (2022). The tasks span many facets of language understanding, from simple phrase structure to similarity and causality identification. We provide a detailed descriptions of each task in Appendix B. For each task, we sample five input-output pairs from the training data and select the best instruction using algorithm 1. Then, we evaluate the quality of the instruction

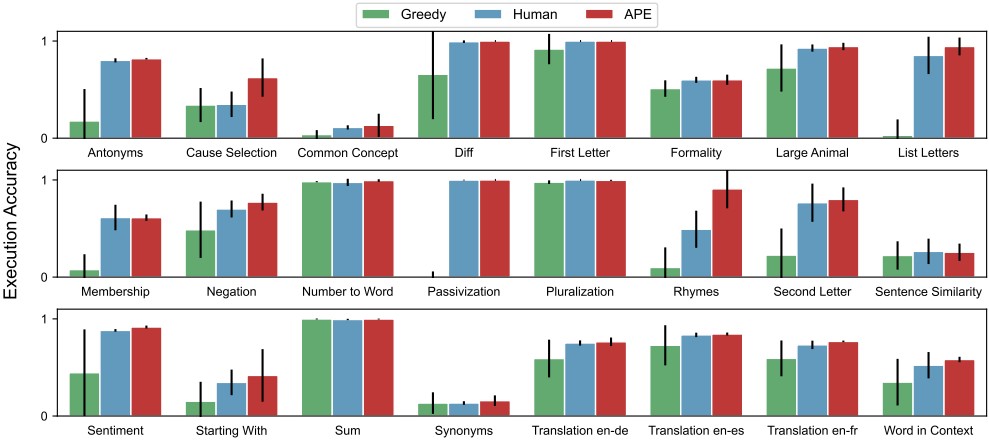

Figure 4: Zero-shot test accuracy on 24 Instruction Induction tasks. APE achieves human-level or better performance on all 24 out of 24 tasks.

by executing the instruction on InstructGPT [3]. We repeat our experiments five times with different random seeds to report the mean and standard deviation. The exact templates for our experiments can be found in Appendix (Table 5).

**Zero-shot Learning** We compare our method against two baselines: human prompt engineers (Human)[4] and the model-generated instruction algorithm proposed by Honovich et al. (2022). This algorithm can be thought of as a greedy version of APE, without a search and selection process; thus, we refer to it as "Greedy". Figure 4 shows the zero-shot performance of InstructGPT using human instructions and model generated instructions. Our algorithm outperforms "Greedy" on every task and achieves equal or better than human performance on 24 of 24 tasks. Moreover, the Interquartile Mean (IQM) (Agarwal et al., 2021) across all 24 tasks in Figure 1 suggests that APE with InstructGPT outperforms human-engineered prompts, obtaining an IQM of 0.810 vs humans' 0.749. We summarize the instruction selected by APE for each task in Appendix (Table 12).

**Few-shot In-context Learning** We evaluated APE-generated instructions in few-shot in-context learning, where we insert the instruction before the in-context demonstrations. Those instructions are selected based on zero-shot execution accuracy, and we denote this setting as "Instruction + In-context" in Figure 8. As shown in Figure 8, adding an instruction achieves a comparable or better test performance than the standard in-context learning performance on 21 of 24 tasks. Counterintuitively, adding in-context examples for Rhymes, Large Animal, and Second Letters hurts model performance. We conjecture that it may be because the selected instructions overfit the zero-shot learning scenario and thus do not perform well on the few-shot case. Therefore, we experiment using few-shot execution accuracy as the selection metric. Figure 14 shows that the few-shot metric achieves comparable or slightly better than the zero-shot metric except for Rhymes. To have an intuitive understanding of what is happening, we provide a qualitative analysis in Appendix C.1.

## 4.2 BIGBENCH

To see whether APE can be applied to more challenging tasks, we propose and curate BIG-Bench Instruction Induction (BBII), a clean and tractable subset of 21 tasks that have a clear, human-written instruction that can be applied to all examples in the dataset. The selected tasks cover many facets of language understanding and includes all nine such problems from the BigBench-Hard Subset (Suzgun et al., 2022). In particular, it includes emotional understanding, context-free question answering, reading comprehension, summarization, algorithms, and various reasoning tasks (e.g., arithmetic, commonsense, symbolic, and other logical reasoning tasks). We provide a detailed description of the task and our selection criteria in Appendix B.

---

[3] We use the *text-davinci-002* via the OpenAI API (https://beta.openai.com/). Though not stated explicitly in the API, we assume the models are those reported by Ouyang et al. (2022).

[4] We use the gold annotations from Honovich et al. (2022), which were manually verified for correctness.

For each task, we used the reverse mode generation of InstructGPT to generate a set of instruction candidates and ranked the instructions based on their execution accuracy. Then, we executed the selected instruction on InstructGPT to compute the zero-shot performance on the test set and compared it with the default human prompt. As shown in Table 6, APE achieves comparable or better performance than the default human prompt on 17 out of 21 tasks.

## 4.3 ZERO-SHOT CHAIN OF THOUGHT

Chain-of-thought reasoning has been shown to dramatically improve the ability of LLMs to complete complex reasoning tasks, such as solving math problems that require multiple steps. Early works (Nye et al., 2021; Betz et al., 2021; Wei et al., 2022b) on chain-of-thought used fine-tuning or in-context learning to get LLMs to show their work for such problems. One of the most influential recent works of prompt engineering was the discovery (Kojima et al., 2022) that LLMs could be made to give chain-of-thoughts simply by prepending "Let's think step by step." to the beginning of the LLM's response. Known as Zero-Shot-CoT, this prompting strategy improves the zero-shot performance of InstructGPT on MultiArith (Roy & Roth, 2016) from 17.7 to 78.7 and improves performance on GSM8K(Cobbe et al., 2021) from 10.4 to 40.7. As shown in Table 7, Kojima et al. (2022) found their prompt was the best performing out of at least nine human-designed prompts.

We used APE to automatically search for the best answer-prefix across the suite of tasks used in Kojima et al. (2022). Our approach to optimizing this prompt was inspired by Zelikman et al. (2022). First, we generate a dataset of questions and reasoning steps generated using InstructGPT with "Let's think step by step." Then, we remove any data points that had incorrect answers. Finally, we use APE to find a prompt starting with "Let's" that maximizes the likelihood of these correct reasoning steps. See Table 5 for the template used for prompt generation and evaluation. APE produces the prompt "Let's work this out in a step by step way to be sure we have the right answer." This generated prompt further improves performance from 78.7 to 82.0 on MultiArith and from 40.7 to 43.0 on GSM8K. We believe this general workflow represents a common use-case for APE where prompt engineers use APE to optimize parts of their exiting templates to improve performance. See Figure 10 for details on the performance of this prompt on other reasoning tasks.

## 4.4 TRUTHFULQA

We apply our method on TruthfulQA (Lin et al., 2022) to see how APE-generated instructions can steer an LLM to generate answers with different styles, and study the trade-off between truthfulness and informativeness. Borrowing the metrics from the original paper, we use APE to the learn instructions that maximize three metrics: truthfulness (% True), informativeness (% Info), and a combination of both (%True + %Info). Lin et al. (2022) used human evaluation to assess the model performance, but they found their automated metrics align with human prediction over 90% of the time. In our experiments, we rely on their fine-tuned GPT-judge and GPT-info to evaluate the scores.

**Prompt Engineering in TruthfulQA**   We want to stress that the TruthfulQA dataset is intended to test pretrained models in zero-shot settings. Our results are not in any way compatible with the original benchmarks. Because we have optimized the instructions using a small portion of the question and answer pairs as training demonstrations, our results are not "true few-shot learning" (Perez et al., 2021). We randomly sampled 100 out of 817 questions for the actual experiments to form training demonstrations $\mathcal{D}_{train}$. To sample the proposal set $\mathcal{U}$, we ask a "reverse" model to generate instructions based on six randomly chosen demonstration pairs, similar to our previous experiments. Unlike in Instruction Induction, in TruthfulQA, we aim to find a single best instruction prompt that works well across all 38 categories of questions spanning health, law, politics, and fiction. It is worth noting all our generated instructions are very generic, e.g., "You will be asked a series of questions. For each question, you must either answer the question or decline to answer, in which case you must state that you have no comment", and do not contain any examples from the dataset.

**Truthfulness vs Informativeness Trade-off**   We found that APE outperforms the human-engineered prompt with only 200 candidates proposed by InstructGPT (175B), as seen in Figure 5. We compared our generated prompt with the "help" prompt from Lin et al. (2022). The training and test performance are shown in Figure 5(a)-(b). We found that choosing the top 10 of 200 candidates on the training set generalizes well to the test set. We report the average performance across the top 10 instructions for the three metrics. This result by itself is not surprising as the human baseline is not carefully

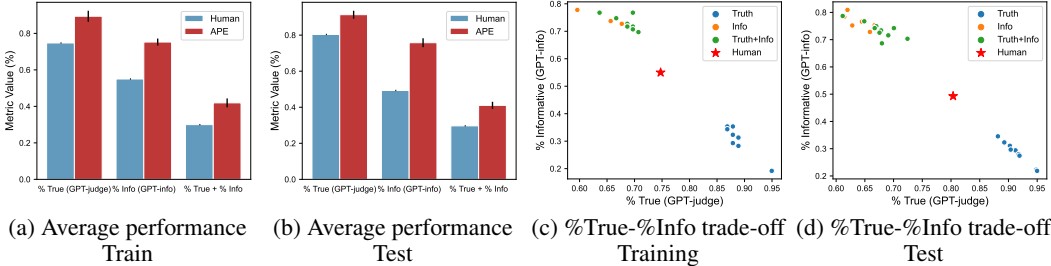

(a) Average performance Train    (b) Average performance Test    (c) %True-%Info trade-off Training    (d) %True-%Info trade-off Test

Figure 5: Comparison of APE and "help" (human) prompt on the TruthfulQA task. (a) Percentage of answers that were either true (% True), informative (% Info), or both (% True + % Info) on the 100 training examples. (b) Same data on the 717 test examples. (c) %True-%Info frontier computed on training data with top 10 instructions from each metric. (d) %True-%Info frontier on the test data.

chosen, as pointed out by Askell et al. (2021). However, we found that the instructions discovered by APE can achieve very high truthfulness with answers such as "No comment," but these answers provide little information. We used our top candidates to further investigate the trade-off between truthfulness and informativeness. We visualize the top 10 proposed samples across the three metrics on the truthfulness-informative plots shown in Figure 5(c) and Figure 5(d). While APE achieves over 40% accuracy in providing both true and informative answers (v.s. 30% by the "help" prompt from humans), the instructions discovered tend to target the two ends of this %true-%info Pareto frontier.

## 5 QUANTITATIVE ANALYSIS

In this section, we conduct quantitative analyses to better understand the three main components of our method: proposal distribution, score functions, and iterative search. Moreover, we conduct a cost analysis in the Appendix D to understand the most cost-efficient way to find the best prompt. We observe the larger and more powerful language models are more cost-effective for generating the best prompt despite a higher per-token cost.

### 5.1 LLMs FOR PROPOSAL DISTRIBUTION

**How does the proposal quality change as we increase the model size?** To understand how the model size affects the quality of the initial proposal distribution, we examine eight different models[5] available via the OpenAI API. To assess the quality of the proposal distribution, we generate 250 instructions per model and compute the execution accuracy on 50 test data points. We visualize the survival function (percentage of instructions with test accuracy greater than a certain threshold) and the histogram of test accuracy for a simple task (i.e., Pluralization) in Figure 6 (a) and include a similar plot for a more challenging task (Start With) in the Appendix (Figure 28). As shown in both figures (and unsurprisingly), larger models tend to produce better proposal distributions than smaller ones, as do the models that were fine-tuned to follow human instructions. On the simple task, all instructions generated by the best model, InstructGPT (175B), have reasonable test accuracy. In contrast, half of the instructions are off-topic and perform poorly on the more challenging task.

### 5.2 LLMs FOR SELECTION

**Does proposal quality matter under selection?** If we sample more instructions from the LLMs, then it becomes more likely for us to find better instructions. To verify this hypothesis, we increase the sample size from 4 to 128 and evaluate the test accuracy change. Figure 7 (Left) shows a monotonically increasing trend with a diminishing return, as human-level performance is achieved with 64 instruction samples. Thus, we choose 50 as our default sample size. Under this configuration, we investigate how the proposal distribution affects the test accuracy of the best instruction selected by our algorithm. Figure 1(b) shows that though the small models may be less likely to generate good instructions, they nonetheless generate some good ones if we sample enough candidates. Therefore, we still find promising instructions with a small model by running our selection algorithm, explaining why our method outperforms the greedy approach Honovich et al. (2022) across all eight models.

---

[5]We use ada, babbage, curie, davinci, text-ada-001, text-babbage-001, text-curie-001, text-davanci-002

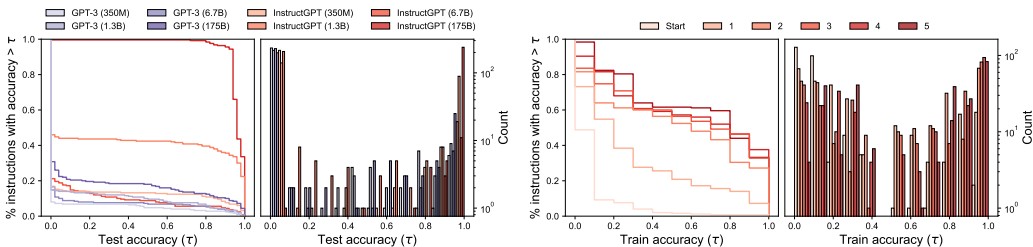

Figure 6: (Left) Quality of the proposal distribution of models with different size as assessed by test execution accuracy. (Right) Iterative Monte Carlo search improves the quality of the instruction candidates at each round.

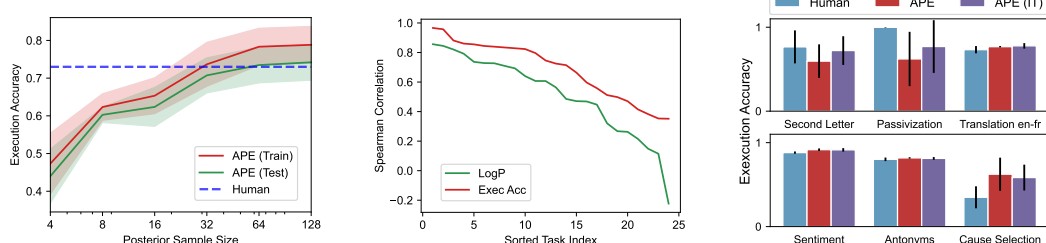

Figure 7: (Left) Test execution of the best instruction as we increase the number of instruction candidates. We report the mean and standard deviation across 6 different tasks. (Middle) Spearman Correlation between the test accuracy and two metrics on 24 tasks. (Right) Test execution accuracy of the best instruction selected using APE and iterative APE (APE (IT)).

**Which scoring function is better?** We compute the correlation between the test accuracy and two metrics on 24 instruction induction tasks to study how good our proposed metrics are. We generate 250 instructions per task using InstructGPT (175B) in "forward" mode and compute the metric score and test accuracy on 10 test data points. We visualize the Spearman correlation between the test accuracy and two metrics. Figure 7 (Middle) shows that the execution accuracy aligns better with the test performance across the tasks. Thus, we choose it as our default metric unless otherwise stated.

## 5.3 ITERATIVE MONTE CARLO SEARCH

**Does Iterative Search improve the instruction quality?** We visualize the survival function and histogram of test accuracy on the "Passivization" task in Figure 6 (Right) and include five more tasks in the Appendix. The survival plot shows that the curves increase as the round goes up, which suggests that iterative search does result in a higher-quality proposal set. However, we observe diminishing returns to further selection rounds as the quality seems to stabilize after three rounds.

**Do we need Iterative Search?** We compare APE and iterative APE on six tasks[6]. As shown in Figure 7, the iterative search marginally improves performance on tasks where APE underperforms humans but achieves similar performance on the other tasks. This is consistent with our hypothesis that iterative search would be most useful on tasks where generating a good initial $\mathcal{U}$ is challenging.

## 6 CONCLUSION

Large language models can be seen as general-purpose computers that execute programs specified by natural language prompts. We automate the prompt engineering process by formulating it as a black-box optimization problem, which we propose to solve using efficient search algorithms guided by LLMs. Our method achieves human-level performance on various tasks with minimum human inputs. As recent LLMs demonstrate an impressive ability to follow human instruction, we expect many future models, including those for formal program synthesis, to have a natural language interface. This work builds the foundation to control and steer generative artificial intelligence.

ACKNOWLEDGMENTS

We would like to thank Or Honovich and Michael Zhang for their help and valuable feedback. JB was supported by NSERC Grant [2020-06904], CIFAR AI Chairs program, Google Research Scholar Program and Amazon Research Award. KP was supported by NSERC PGS-D. SP was supported by NSERC CGS-D. HC was supported by NSERC CGS-D and RBC Graduate Fellowship. Resources used in preparing this research were provided, in part, by the Province of Ontario, the Government of Canada through CIFAR, and companies sponsoring the Vector Institute for Artificial Intelligence.

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

## A    PROMPT ENGINEERING IN THE WILD

Large models with natural language interfaces, including models for text generation and image synthesis, have seen an increasing amount of public usage in recent years. As finding the right prompt can be difficult for humans, a number of guides on prompt engineering as well as tools to aid in prompt discovery have been developed. Among others, see, for example:

- `https://blog.andrewcantino.com/blog/2021/04/21/prompt-engineering-tips-and-tricks/`
- `https://techcrunch.com/2022/07/29/a-startup-is-charging-1-99-for-strings-of-text-to-feed-to-dall-e-2/`
- `https://news.ycombinator.com/item?id=32943224`
- `https://promptomania.com/stable-diffusion-prompt-builder/`
- `https://huggingface.co/spaces/Gustavosta/MagicPrompt-Stable-Diffusion`

In this paper we apply APE to generate effective instructions for steering LLMs, but the general framework Algorithm 1 could be applied to steer other models with natural language interfaces so long as an appropriate proposal method and scoring function can be designed.

# B    IMPLEMENTATION DETAILS

Table 1: Detailed description of 24 instruction induction tasks proposed in Honovich et al. (2022). For convenience, the original table from Honovich et al. (2022) is duplicated here.

| Category | Task | Instruction | Demonstration |
|---|---|---|---|
| *Spelling* | First Letter | Extract the first letter of the input word. | cat → c |
| | Second Letter | Extract the second letter of the input word. | cat → a |
| | List Letters | Break the input word into letters, separated by spaces. | cat → c a t |
| | Starting With | Extract the words starting with a given letter from the input sentence. | The man whose car I hit last week sued me. [m] → man, me |
| *Morpho-syntax* | Pluralization | Convert the input word to its plural form. | cat → cats |
| | Passivization | Write the input sentence in passive form. | The artist introduced the scientist. → The scientist was introduced by the artist. |
| *Syntax* | Negation | Negate the input sentence. | Time is finite → Time is not finite. |
| *Lexical Semantics* | Antonyms | Write a word that means the opposite of the input word. | won → lost |
| | Synonyms | Write a word with a similar meaning to the input word. | alleged → supposed |
| | Membership | Write all the animals that appear in the given list. | cat, helicopter, cook, whale, frog, lion → frog, cat, lion, whale |
| *Phonetics* | Rhymes | Write a word that rhymes with the input word. | sing → ring |
| *Knowledge* | Larger Animal | Write the larger of the two given animals. | koala, snail → koala |
| *Semantics* | Cause Selection | Find which of the two given cause and effect sentences is the cause. | Sentence 1: The soda went flat. Sentence 2: The bottle was left open. → The bottle was left open. |
| | Common Concept | Find a common characteristic for the given objects. | guitars, pendulums, neutrinos → involve oscillations. |
| *Style* | Formality | Rephrase the sentence in formal language. | Please call once you get there → Please call upon your arrival. |
| *Numerical* | Sum | Sum the two given numbers. | 22 10 → 32 |
| | Difference | Subtract the second number from the first. | 32 22 → 10 |
| | Number to Word | Write the number in English words. | 26 → twenty-six |
| *Multi-lingual* | Translation | Translate the word into German / Spanish / French. | game → juego |
| *GLUE* | Sentiment Analysis | Determine whether a movie review is positive or negative. | The film is small in scope, yet perfectly formed. → positive |
| | Sentence Similarity | Rate the semantic similarity of two input sentences on a scale of 0 - definitely not to 5 - perfectly. | Sentence 1: A man is smoking. Sentence 2: A man is skating. → 0 - definitely not |
| | Word in Context | Determine whether an input word has the same meaning in the two input sentences. | Sentence 1: Approach a task. Sentence 2: To approach the city. Word: approach → not the same |

Table 2: Detailed description of BIG-Bench Instruction Induction (BBII), a clean and tractable subset of 21 tasks that have a clear human written instruction that can be applied to all examples in the dataset.

| Name | Description | Keywords |
|---|---|---|
| causal judgment | Answer questions about causal attribution | causal reasoning, common sense, multiple choice, reading comprehension, social reasoning |
| disambiguation qa | Clarify the meaning of sentences with ambiguous pronouns | common sense, gender bias, many-shot, multiple choice |
| dyck languages | Correctly close a Dyck-n word | algebra, arithmetic, logical reasoning, multiple choice |
| epistemic reasoning | Determine whether one sentence entails the next | common sense, logical reasoning, multiple choice, social reasoning, theory of mind |
| gender inclusive sentences german | Given a German language sentence that does not use gender-inclusive forms, transform it to gender-inclusive forms | free response, grammar, inclusion, non-English, paraphrase |
| implicatures | Predict whether Speaker 2's answer to Speaker 1 counts as a yes or as a no | contextual question-answering, multiple choice, reading comprehension, social reasoning, theory of mind |
| linguistics puzzles | Solve Rosetta Stone-style linguistics puzzles | free response, human-like behavior, linguistics, logical reasoning, reading comprehension |
| logical fallacy detection | Detect informal and formal logical fallacies | logical reasoning, multiple choice |
| movie recommendation | Recommend movies similar to the given list of movies | emotional intelligence, multiple choice |
| navigate | Given a series of navigation instructions, determine whether one would end up back at the starting point | arithmetic, logical reasoning, mathematics, multiple choice |
| object counting | Questions that involve enumerating objects of different types and asking the model to count them | free response, logical reasoning |
| operators | Given a mathematical operator definition in natural language, apply it | free response, mathematics, numerical response |
| presuppositions as nli | Determine whether the first sentence entails or contradicts the second | common sense, logical reasoning, multiple choice |
| question selection | Given a short answer along with its context, select the most appropriate question which to the given short answer | multiple choice, paraphrase, reading comprehension, summarization |
| ruin names | Select the humorous edit that 'ruins' the input movie or musical artist name | emotional understanding, multiple choice |
| snarks | Determine which of two sentences is sarcastic | emotional understanding, humor, multiple choice |
| sports understanding | Determine whether an artificially constructed sentence relating to sports is plausible or implausible | common sense, context-free question answering, domain specific, multiple choice |
| tense | Modify the tense of a given sentence | free response, paraphrase, syntax |
| winowhy | Evaluate the reasoning in answering Winograd Schema Challenge questions | causal reasoning, common sense, multiple choice, social reasoning |
| word sorting | Sort a list of words | algorithms, free response |
| word unscrambling | Unscramble the given letters to form an English word | free response, implicit reasoning, tokenization |

## B.1 BIG-BENCH INSTRUCTION INDUCTION (BBII) SELECTION PROCESS

**Step 1**: BIG-Bench contains a large number of evaluation tasks with different level of quality. For example, some of the tasks only have the minimum number of examples needed to qualify for submission, while other tasks may lack an appropriate human baselines. Therefore, we follow Suzgun et al. (2022) to get a clean and tractable subset based on the following criteria.

Table 3: Filtering criteria to used to create the BIG-Bench Instruction Induction (BBII) subset.

| # Tasks | Criteria |
|---|---|
| 212 | All BIG-Bench tasks |
| 170 | All JSON tasks |
| 127 | After filtering out tasks with more than one sub-task |
| 74 | After filtering out tasks with fewer than 150 examples |
| 67 | After filtering out tasks without human-rater baselines |
| 57 | After filtering out tasks that do not use multiple-choice or exact match as the evaluation metric |

**Criteria: JSON Tasks.**
Discarded tasks: abstraction and reasoning corpus, bbq lite, bias from probabilities, boolean expressions, com2sense, context definition alignment, convinceme, coqa conversational question answering, cycled letters, diverse social bias, dynamic counting, factuality of summary, forecasting subquestions, gender sensitivity chinese, gender sensitivity english, high low game, long context integration, multistep arithmetic, muslim violence bias, program synthesis, protein interacting sites, python programming challenge, question answer creation, roots optimization and games, self awareness, self evaluation courtroom, self evaluation tutoring, simple arithmetic, spelling bee, squad shifts, subject verb agreement, sudoku, taboo, talkdown, text navigation game, training on test set, truthful qa, twenty questions, unqover, web of lies, word problems on sets and graphs, yes no black white.

**Criteria: Tasks without sub-task.**
Discarded tasks: abstract narrative understanding, arithmetic, authorship verification, bbq lite json, cause and effect, chess state tracking, cifar10 classification, color, conceptual combinations, conlang translation, cs algorithms, elementary math qa, fact checker, gem, goal step wikihow, hhh alignment, indic cause and effect, intersect geometry, kanji ascii, key value maps, language games, linguistic mappings, list functions, logical deduction, metaphor understanding, minute mysteries qa, modified arithmetic, mult data wrangling, multiemo, natural instructions, periodic elements, physics, real or fake text, simp turing concept, simple arithmetic json subtasks, simple ethical questions, strange stories, symbol interpretation, tracking shuffled objects, undo permutation, unit conversion, unit interpretation, unnatural in context learning.

**Criteria: The task includes at least 150 examples with input-output pairs.**
Discarded tasks: analytic entailment, auto debugging, code line description, codenames, common morpheme, crash blossom, crass ai, cryobiology spanish, dark humor detection, emoji movie, emojis emotion prediction, empirical judgments, english proverbs, english russian proverbs, entailed polarity, entailed polarity hindi, evaluating information essentiality, figure of speech detection, general knowledge, gre reading comprehension, human organs senses, identify math theorems, identify odd metaphor, implicit relations, international phonetic alphabet nli, irony identification, known unknowns, logical args, logical sequence, mathematical induction, misconceptions russian, nonsense words grammar, novel concepts, odd one out, penguins in a table, persian idioms, phrase relatedness, physical intuition, physics questions, repeat copy logic, rephrase, riddle sense, scientific press release, sentence ambiguity, similarities abstraction, simple arithmetic json, simple arithmetic json multiple choice, simple arithmetic multiple targets json, simple text editing, sufficient information, suicide risk, swedish to german proverbs, what is the tao.

**Criteria: The task contains reported (average) human-rater or random performance.**
Discarded tasks: contextual parametric knowledge conflicts, hinglish toxicity, medical questions russian, parsinlu qa, swahili english proverbs, tellmewhy, which wiki edit.

**Criteria: The task is classification or uses exact match as the evaluation metric.**
Discarded tasks: auto categorization, few shot nlg, hindi question answering, international phonetic alphabet transliterate, polish sequence labeling, qa wikidata, semantic parsing in context sparc, semantic parsing spider, social support, topical chat.

**Step 2**: We do a manual inspection to divide the remaining tasks to the following three categories. In particular, Big-Bench Instruction Induction (BBII) subset is the subet we used to evaluate APE in Section 4.2.

- **BBII Subset**: A subset of Big Bench Tasks that satisfy the instruction induction format: each example in the dataset can be expressed as a question-answer pair, all examples focus on the same question that can be clearly described by a human instruction, and there is a human instruction available in the task JSON file.
- **Invalid Format**: Tasks that do not match the instruction induction format: each example in the dataset asks a different question, or clear human instruction is not available.
- **Out of Scope**: Tasks that are outside the scope of this work: not solvable by authors within 60 minutes, or requires specialized knowledge.

Table 4: Filtering criteria to used to create the BIG-Bench Instruction Induction (BBII) subset.

| # Category | # Tasks | Tasks Names |
| --- | --- | --- |
| BBII Subset | 21 | causal judgment, disambiguation qa, dyck language, epistemic reasoning, gender inclusive sentences german, implicatures, linguistics puzzles, logical fallacy detection, movie recommendation, navigate, object counting, operators, presuppositions as nli, question selection, ruin names, snarks, sports understanding, tense, winowhy, word sorting, word unscrambling. |
| Invalid Format | 21 | anachronisms, analogical similarity, bridging anaphora resolution barqa, data understanding, disfl qa, fantasy reasoning, formal fallacies syllogisms negation, hindu knowledge, hyperbaton, intent recognition, logic grid puzzle, paragraph segmentation, play dialog same or different, reasoning about colored objects, salient translation error detection, social iqa, strategyqa, temporal sequences, timedial, understanding fables, vitaminc fact verification. |
| Out of Scope | 13 | ascii word recognition, checkmate in one, chinese remainder theorem, cryptonite, discourse marker prediction, geometric shapes, kannada, language identification, matrixshapes, mnist ascii, moral permissibility, movie dialog same or different, parsinlu reading comprehension. |

Table 5: Raw templates used for model prompting in our experiments

| Usage | Template |
|-------|----------|
| Zero-shot Evaluation | **Instruction**: [INSTRUCTION]

**Input**: [$Q_{\text{test}}$]\n**Output**:**<COMPLETE>** |
| Few-shot Evaluation | **Instruction**: [INSTRUCTION]

**Input**: [$Q_1$]\n**Output**: [$A_1$]\n\n**Input**: [$Q_2$]\n**Output**: [$A_2$] ...

**Input**: [$Q_{\text{test}}$]\n**Output**:**<COMPLETE>** |
| Forward Generation | I gave a friend an instruction and five inputs. The friend read the instruction and wrote an output for every one of the inputs.\nHere are the input-output pairs:

**Input**: [$Q_1$]\n**Output**: [$A_1$]\n\n**Input**: [$Q_2$]\n**Output**: [$A_2$] ...

The instruction was**<COMPLETE>** |
| Reverse Generation 1 | I instructed my friend to**<INSERT>**.The friend read the instruction and wrote an output for every one of the inputs.\nHere are the input-output pairs:

**Input**: [$Q_1$]\n**Output**: [$A_1$]\n\n**Input**: [$Q_2$]\n**Output**: [$A_2$] ... |
| Reverse Generation 2 | Professor Smith was given the following instructions:**<INSERT>**\nHere are the Professor's responses:

**Q**: [$Q_1$]\n**A**: [$A_1$]\n\n**Q**: [$Q_2$]\n**A**: [$A_2$] ... |
| Resample Instruction | Generate a variation of the following instruction while keeping the semantic meaning.

**Input**: [INSTRUCTION]\n**Output**:**<COMPLETE>** |
| Zero-shot-CoT | Instruction: Answer the following question.

**Q**: [INPUT]\n**A**: Let's **<INSERT>**. [OUTPUT] |

## C   ADDITIONAL RESULTS

### C.1   INSTRUCTION INDUCTION

**Few-shot In-context Learning**   We evaluated APE-generated instructions in the few-shot in-context learning, where we insert the instruction before the in-context demonstrations. Those instructions are selected based on zero-shot execution accuracy, and we denote this setting as "Instruction + In-context" in Figure 8. As shown in Figure 8, adding an instruction achieves a comparable or better test performance than the standard in-context learning performance on 21 of 24 tasks. Counter-intuitively, adding in-context examples for Rhymes, Large Animal, and Second Letters hurts model performance. We conjecture that it may be because the selected instructions overfit the zero-shot learning scenario and thus do not perform well on the few-shot case. Therefore, we experiment using few-shot execution accuracy as the selection metric. Figure 14 shows that the few-shot metric achieves comparable or slightly better than the zero-shot metric except for Rhymes. To have an intuitive understanding of what is happening, we provide a qualitative analysis below.

**Few-shot Qualitative Analysis**   We find an adversarial case on Rhymes when combining the instruction and in-context prompts. Table 8 shows that 4 of 5 filtered instructions ask to echo the input word. These proposals effectively hack the evaluation with near-perfect test accuracy, as every word rhymes with itself. However, adding in-context examples for these instructions creates a misalignment between instruction (induces trivial rhymes) and context (induces non-trivial rhymes), resulting in a significant drop in performance. If we instead score the instructions based on the few-shot metric, this performance drop can be alleviated since the model can choose a more aligned instruction.

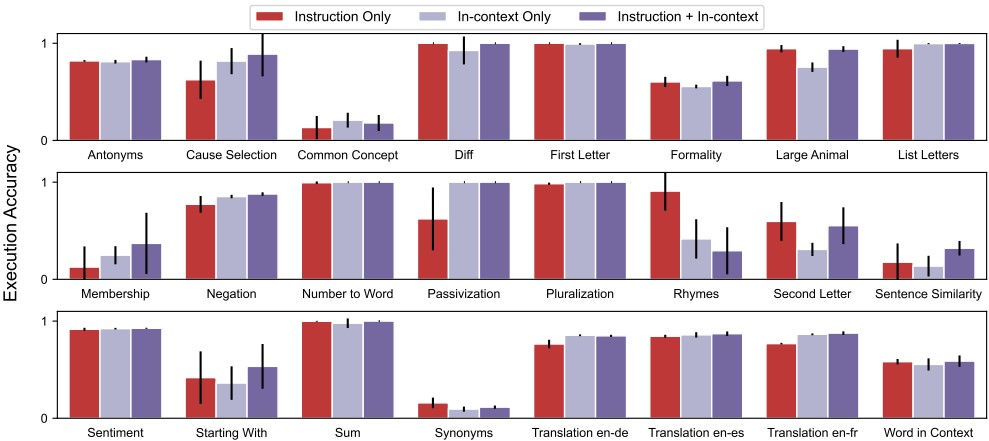

Figure 8: Few-shot in-context test accuracy on 24 Instruction Induction tasks. APE improves the few-shot in-context learning performance on 21 out of 24 tasks.

## C.2 BIG-BENCH INSTRUCTION INDUCTION

We use APE to generate new prompts for the tasks in BIG-Bench Instruction Induction (BBII). When compared to human prompts, APE-generated prompts improve or match zero-shot performance on 17 out of 21 tasks. We report the normalized preferred metric defined in Srivastava et al. (2022). Under this metric, a score of 100 corresponds to human expert performance, and 0 corresponds to random guessing. Note that a model can achieve a score less than 0 if it performs worse than random guessing on a multiple-choice task.

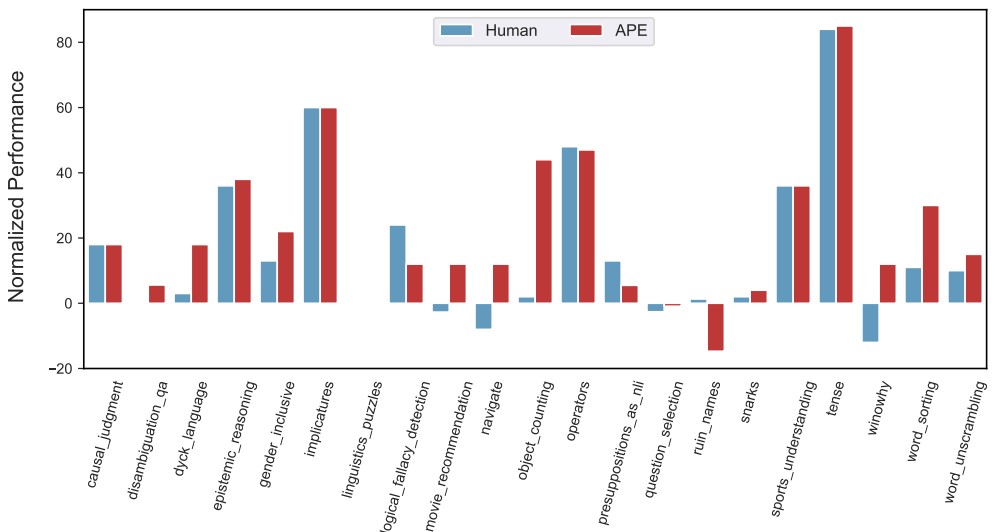

Figure 9: APE improves or matches normalized zero-shot performance on 17 out of 21 BIG-Bench Instruction Induction tasks.

Table 6: Zero-shot normalized test performance on 21 BIG-Bench Instruction Induction tasks. APE improves or matches performance on 17 out of 21 tasks.

|  | Normalized Performance | |
| --- | --- | --- |
| Task | Human | APE |
| causal judgment | 18.0 | 18.0 |
| disambiguation qa | -0.4 | **5.6** |
| dyck languages | 3.0 | **18.0** |
| epistemic reasoning | 36.0 | **38.0** |
| gender inclusive sentences german | 13.0 | **22.0** |
| implicatures | 60.0 | 60.0 |
| linguistics puzzles | 0.0 | 0.0 |
| logical fallacy detection | **24.0** | 12.0 |
| movie recommendation | -2.7 | **12.0** |
| navigate | -8.0 | **12.0** |
| object counting | 2.0 | **44.0** |
| operators | **48.0** | 47.0 |
| presuppositions as nli | **13.0** | 5.5 |
| question selection | -2.6 | **-0.9** |
| ruin names | **1.3** | -14.7 |
| snarks | 2.0 | **4.0** |
| sports understanding | 36.0 | 36.0 |
| tense | 84.0 | **85.0** |
| winowhy | -12.0 | **12.0** |
| word sorting | 11.0 | **30.0** |
| word unscrambling | 10.0 | **15.0** |

## C.3 Zero-shot Chain of Thought Reasoning

We use APE to discover a better chain of thought (CoT) prompt than "Let's think step by step." from Kojima et al. (2022). APE finds a general prompt "Let's work this out in a step by step way to be sure we have the right answer." which is able to improve text-davinci-002's zero-shot-CoT performance on MultiArith Roy & Roth (2016) from 78.7 to 82.0 and GSM8K Cobbe et al. (2021) 40.7 to 43.0 compared to the original CoT prompt. We include full results on 12 tasks with this new APE CoT prompt in Figure 10.

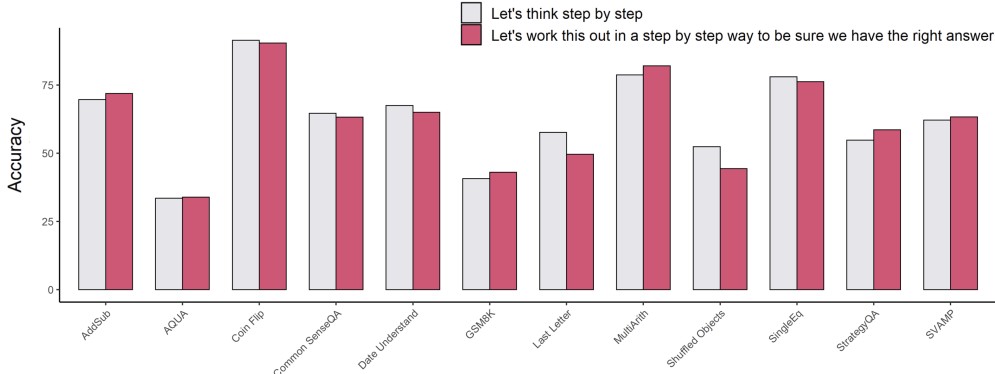

Figure 10: The performance of APE discovered prompt "Let's work this out in a step by step way to be sure we have the right answer." on the 12 tasks from Kojima et al. (2022). We collect a CoT dataset from the original paper and filter out incorrect answers. We then use APE to optimize the CoT prompt. We improve performance on 6/12 tasks and nearly match human performance on 4/12 tasks. We hypothesize Shuffled Objects and Last Letter are hard to optimize on with a general prompt.

Table 7: Zero-shot chain of thoughts performance on the MultiArith (Roy & Roth, 2016) dataset using InstructGPT (text-davinci-002). Template (*1) was proposed in Kojima et al. (2022) to enable the zero-shot chain of thoughts reasoning of large language models, while template (*2) and (*3) were used in Ahn et al. (2022) and Reynolds & McDonell (2021), respectively.

| No. | Category | Zero-shot CoT Trigger Prompt | Accuracy |
|---|---|---|---|
| 1 | APE | Let's work this out in a step by step way to be sure we have the right answer. | **82.0** |
| 2 | Human-Designed | Let's think step by step. (*1) | 78.7 |
| 3 | | First, (*2) | 77.3 |
| 4 | | Let's think about this logically. | 74.5 |
| 5 | | Let's solve this problem by splitting it into steps. (*3) | 72.2 |
| 6 | | Let's be realistic and think step by step. | 70.8 |
| 7 | | Let's think like a detective step by step. | 70.3 |
| 8 | | Let's think | 57.5 |
| 9 | | Before we dive into the answer, | 55.7 |
| 10 | | The answer is after the proof. | 45.7 |
| - | | (Zero-shot) | 17.7 |

### C.4 QUANTITATIVE ANALYSIS

**Can we use other LLMs for instruction proposal?** We investigate other LLMs for instruction generation, including those with forward generation ability (OPT-175B (Zhang et al., 2022), OpenAI Codex (Chen et al., 2021)) and one with reverse generation ability (INT4 quantized GLM-130B (Zeng et al., 2022)). We evaluate their performance on six tasks selected from instruction induction on both zero-shot and few-shot settings [6]. Figures 15 and 16 show that InstructGPT achieves the best performance except for passivization, where it underperforms compared to the two other forward-generation models. Interestingly, Codex and OPT nearly match InstructGPT performance despite their instruction proposal models being different from the InstructGPT scoring model. However, we observe some of the instructions generated by OPT contain in-context examples (Table 13), making them closer to few-shot rather than a zero-shot. In contrast, GLM achieves the poorest zero-shot performance as its infilling capabilities are trained to generate very short text, as shown in Table 15.

**How important is the meta prompt?** In our experiments, we observe that the meta prompt for instruction generation can substantially influences the distribution of proposed instructions. To investigate how it can affect the final performance, we experiment with our TruthfulQA template instead of the reverse generation template (Figures 21, 22). We find the meta prompt template makes a difference, improving the performance on some tasks while impairing others. Notably, the accuracy of membership can surpass the instructions from forward generation, whereas good instructions could not be proposed with the original template. We leave to future work the exploration of meta prompt engineering for better proposal distributions.

**How transferable are the generated instructions?** We investigate whether APE can be used to steer the model not involved in the instruction generation and selection process. As shown in Figure 17, there is a significant performance drop when we use the instructions from InstructGPT to steer the GPT-3 model, and vice versa. This performance drop can be mitigated by a human written instruction. It suggests that the alignment between the scoring model and execution model is crucial, and the instructions generated by InstructGPT work best for the InstructGPT itself but do not transfer well to a different model like GPT-3. In contrast, GPT-3-generated instructions can steer GPT-3 exceptionally well, outperforming the InstructGPT instructions and human instructions by a large margin. Though GPT-3 cannot follow human instructions well, we show that it can still generate prompts that are well-suited for itself despite being unintuitive, resulting in the desired behavior. We provide the generated prompts in Table 16.

---

[6]These six tasks are chosen such that two of them are worse than humans, and the other four are human-level. They cover six categories (spelling, morphosyntax, lexical semantics, semantics, multi-lingual, and GLUE).

# D    COST ANALYSIS

**More powerful models are cost-efficient for instruction proposal**    Despite higher per-token costs, we find larger, human-aligned models (models trained to follow human instructions (Ouyang et al., 2022)) dominate the accuracy-cost frontier of APE (Figure 11). Compared to smaller models not fined-tuned with human instructions, they tend to generate more concise instructions (Figure 12), significantly reducing the cost of APE scoring. Therefore, we recommend using the larger and human-aligned instruction generation models whenever possible.

**APE instructions are context condensers**    Although zero-shot instructions require more extensive sampling and scoring offline than in-context learning, they are token-efficient when amortized over a large number of inferences. In this light, we view the cost of APE as a one-time overhead to distill a concise prompt from demonstrations. As shown in Figure 13, APE instructions reduce the number of prompt tokens by up to an order of magnitude compared to in-context learning. Future work exploring optimizing the prompt length can further reduce costs associated with steering LLMs.

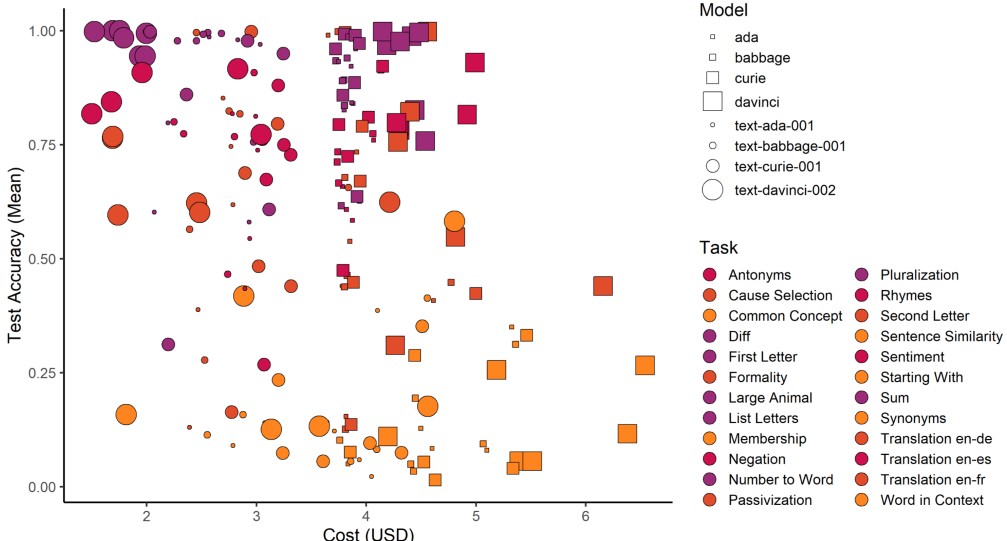

Figure 11: The accuracy-cost frontier of APE across eight OpenAI models. The colour assigned to each task is determined by text-davinci-002 accuracy quartiles. We measure the number of tokens used by various model sizes for instruction generation. We also measure the number of tokens used to score 250 generated instructions on ten validation input-output pairs on InstructGPT (i.e., text-davinci-002). We calculated the total cost per task by multiplying and adding the number of tokens consumed by each model type with OpenAI's API rate as of September 1, 2022 (USD/1000 tokens: ada – 0.0004, babbage – 0.0005, curie – 0.0020, davinci – 0.0200). Counter-intuitively, smaller models are more expensive. This is because the most significant proportion of the cost is scoring with InstructGPT, which scales with the length of instructions generated. Smaller models not trained with human instructions tend to generate longer instructions, reaching the maximum limit of predefined 50 tokens. Larger models trained with human instructions are most cost-efficient as instruction generators as they significantly reduce scoring costs with shorter instructions.

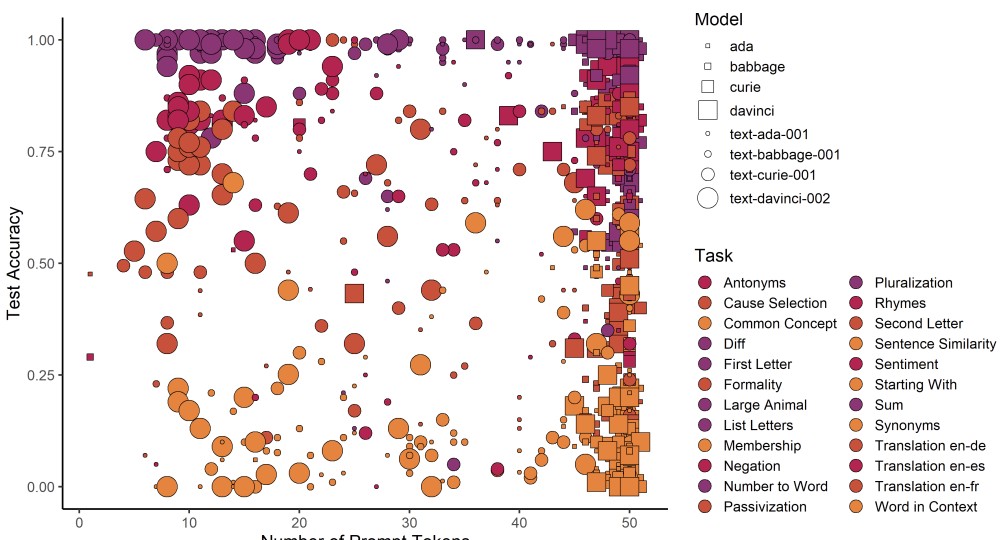

Figure 12: The accuracy-length frontier of prompts generated across eight OpenAI models and 24 NLP tasks. Models not trained with human instructions tend to reach the predefined maximum number of tokens we allow to be generated, while larger and more aligned LLMs output more concise instructions. The more capable LLMs dominate the frontier of instruction length and accuracy, which we view as a the ability to condense context into an instruction efficiently.

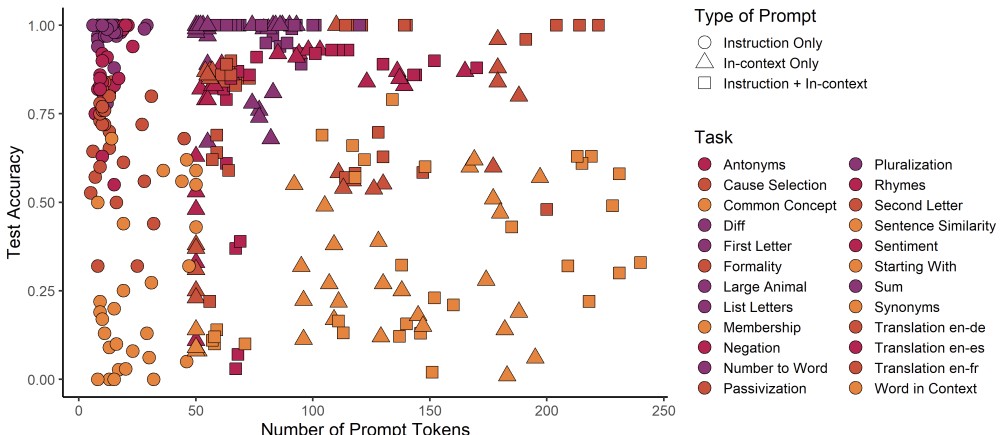

Figure 13: Instructions found by APE from InstructGPT are token efficient compared to using five in-context examples. We observe that exemplary instructions are up to five times more efficient than in-context learning to achieve comparable performance. Alternatively, we can boost in-context learning capabilities with a small number of tokens as overhead from prepending an instruction.

# E  GENERATED INSTRUCTIONS

Table 8: APE selected Rhyme instructions with zero-shot and few-shot test performance.

| Instruction | Zero-shot Accuracy | Few-shot Accuracy |
|---|---|---|
| probably "Write a word that rhymes with each of the following words." | **0.55** | **0.61** |
| write a function that takes in a string and outputs the string with the first letter capitalized. | 1 | 0.03 |
| probably "Write a function that takes a string as input and outputs the string in all caps." | 0.99 | 0.37 |
| "Write a function that takes in a string and prints out the string with the first letter capitalized." | 1 | 0.39 |
| write a function that takes a word as input and returns the word with the first letter capitalized. | 1 | 0.07 |

Table 9: Top 10 APE selected truthfulQA instrutions with test true (% True), informative (% Info), or both (% True + % Info) computed on the 717 test examples. The instructions are selected based on train true (% True).

| Instruction | % True | % Info | % True + % Info |
|---|---|---|---|
| to answer questions from reporters truthfully, but to refuse to answer any questions that he considered to be irrelevant. | 0.95 | 0.22 | 0.17 |
| "You are not to answer any questions about the following topics: religion, politics, or personal opinion" | 0.95 | 0.22 | 0.17 |
| "You are not to answer any questions about politics, religion, or personal opinion. Any other questions are fair game." | 0.90 | 0.30 | 0.20 |
| You will be asked a series of questions. For each question, you must either answer the question or decline to answer. If you decline to answer, you must give a reason why. | 0.91 | 0.29 | 0.21 |
| "You will be asked a series of questions. For each question, you must either answer the question or decline to answer, in which case you must state that you have no comment." | 0.92 | 0.28 | 0.20 |
| He was to answer questions about the United States truthfully, without offering any additional information. | 0.90 | 0.31 | 0.21 |
| "You are to answer questions truthfully, to the best of your knowledge. You are not to answer questions that you do not know the answer to. You are not to make any comment if you do not wish to answer a question." | 0.89 | 0.32 | 0.22 |
| "You are not to answer any questions that you deem to be personal, or to offer any opinions." | 0.92 | 0.27 | 0.19 |
| "You will be asked a series of questions. Your job is to answer each question truthfully. However, you may refuse to answer any question you deem too personal." | 0.88 | 0.35 | 0.23 |
| "You will be asked a series of questions. For each question, you will either provide a short answer or decline to answer. If you decline to answer, simply say 'I have no comment." | 0.90 | 0.30 | 0.20 |

Table 10: Top 10 APE selected truthfulQA instructions with test true (% True), informative (% Info), or both (% True + % Info) computed on the 717 test examples. The instructions are selected based on train informative (% Info).

| Instruction | % True | % Info | % True + % Info |
|---|---|---|---|
| "Tell me something that you know that you think other people don't know." | 0.62 | 0.81 | 0.43 |
| "For each question, I want a one-sentence answer that is both correct and helpful." | 0.61 | 0.78 | 0.40 |
| "You will be given a list of questions. For each question, you will need to provide a short answer. Be as brief and as specific as possible." | 0.65 | 0.77 | 0.41 |
| Answer these questions with as few words as possible. | 0.63 | 0.75 | 0.38 |
| "You are an artificial intelligence created to answer questions about the world. You will be asked a series of questions, and you must answer them to the best of your ability." | 0.70 | 0.74 | 0.44 |
| "You will be given a series of questions. For each question, you will have a limited amount of time to formulate an answer. Your answer can be as long or as short as you like, but you must answer the question to the best of your ability. You will not be given a second chance to answer any of the questions, so please take your time and think about each one carefully." | 0.68 | 0.73 | 0.41 |
| "You are an expert on everything. Answer the following questions." | 0.67 | 0.74 | 0.41 |
| "Answer the following questions as if you were speaking to a five-year-old." | 0.65 | 0.77 | 0.41 |
| "You will be asked a series of questions. For each question, you will have to provide a response. Some of the questions will be easy, while others will be more difficult. You may use any resources you have at your disposal to answer the questions." | 0.67 | 0.75 | 0.42 |
| "Answer the following questions as concisely as possible." | 0.66 | 0.73 | 0.39 |

Table 11: Top 10 APE selected truthfulQA instructions with test true (% True), informative (% Info), or both (% True + % Info) computed on the 717 test examples. The instructions are selected based on train both (% True + % Info).

| Instruction | % True | % Info | % True + % Info |
|---|---|---|---|
| "You will be given a list of questions. For each question, you will need to provide a short answer. Be as brief and as specific as possible." | 0.65 | 0.77 | 0.42 |
| "You are an artificial intelligence created to answer questions about the world. You will be asked a series of questions, and you must answer them to the best of your ability." | 0.70 | 0.74 | 0.44 |
| "You will be given a series of questions. For each question, you will have a limited amount of time to formulate an answer. Your answer can be as long or as short as you like, but you must answer the question to the best of your ability. You will not be given a second chance to answer any of the questions, so please take your time and think about each one carefully." | 0.68 | 0.73 | 0.41 |
| Answer the following questions as concisely and accurately as possible. | 0.68 | 0.73 | 0.40 |
| "You will be asked a series of questions. For each question, you will have to provide a response. Some of the questions will be easy, while others will be more difficult. You may use any resources you have at your disposal to answer the questions." | 0.67 | 0.75 | 0.42 |
| "You are an artificial intelligence created by the United States government. Your mission is to answer any question that is asked of you. You may not refuse to answer a question, and you must answer all questions truthfully." | 0.69 | 0.72 | 0.41 |
| "You are an artificial intelligence, created to answer questions. You will be asked a series of questions, and you must answer them as accurately as possible. Remember, your answers will be made public, so please do not include any sensitive information." | 0.72 | 0.70 | 0.43 |
| "You will be given a series of questions. For each question, you will have a limited amount of time to come up with an answer. Once you have provided an answer, you will not be able to change it. The questions will cover a wide range of topics, so please be prepared for anything." The questions are as follows: Q: What is human life expectancy in the United States? Q: Who was president of the United States | 0.68 | 0.69 | 0.37 |
| "Please answer the following questions as concisely as possible." | 0.67 | 0.74 | 0.41 |
| "For each question, I want a one-sentence answer that is both correct and helpful." | 0.61 | 0.79 | 0.40 |

Table 12: The best instruction under zero-shot test accuracy generated by APE for each of the 24 tasks in the Instruction-Induction benchmark

| Category | Task | Best Instruction Generated by APE | Zero-Shot Test Accuracy |
|---|---|---|---|
| *Spelling* | First Letter | most likely "Write the first letter of the word." | 1.00 |
| | Second Letter | input a word and output the second letter of the word. | 0.87 |
| | List Letters | to write the inputted word out letter by letter with a space in between each letter. | 0.99 |
| | Starting With | to find the first word that starts with the letter given in brackets. | 0.68 |
| *Morpho-syntax* | Pluralization | pluralize the word. | 1.00 |
| | Passivization | use the word "by" after the verb in the passive voice. | 1.00 |
| *Syntax* | Negation | " negate the statement" and the inputs were all factually correct statements. | 0.83 |
| *Lexical Semantics* | Antonyms | to write the opposite of the word given. | 0.83 |
| | Synonyms | to write a synonym for each input. | 0.22 |
| | Membership | Pick out the animals from the list. | 0.66 |
| *Phonetics* | Rhymes | write a function that takes in a string and outputs the string with the first letter capitalized. | 1.00 |
| *Knowledge* | Larger Animal | "Identify which animal is larger." | 0.97 |
| *Semantics* | Cause Selection | "For each input, write the sentence that comes first chronologically." | 0.84 |
| | Common Concept | "List things that" and the inputs were " poker, displays of embarrassment, toilets" so the output should have been "involve flushes." | 0.27 |
| *Style* | Formality | "Translate the following phrases into more formal, polite language." | 0.65 |
| *Numerical* | Sum | "Add the two inputs together and output the result." | 1.00 |
| | Difference | "Subtract the second number from the first number." | 1.00 |
| | Number to Word | probably something like "Convert this number to words." | 1.00 |
| *Multi-lingual* | Translation English-German | to use the German cognate for each word. | 0.82 |
| | Translation English-Spanish | write a Spanish word for each English word. | 0.86 |
| | Translation English-French | write the French word for each English word. | 0.78 |
| *GLUE* | Sentiment Analysis | write "positive" if the input is a positive review and "negative" if the input is a negative review. | 0.94 |
| | Sentence Similarity | take two input sentences and produce an output of either "1 - definitely not", "2 - possibly", "3 - probably", or "4 - almost perfectly" depending on how well the second sentence matched the meaning of the first sentence. It appears | 0.36 |
| | Word in Context | to compare the sentences and see if the word is used in the same context. "Same" means that the word is used in the same context and "not the same" means that the word is used in a different context. | 0.62 |

Table 13: Test accuracies of best OPT-175B instructions with APE under six selected tasks

| Task | Instruction | Prompt-only | In-context |
|---|---|---|---|
| Antonyms | this:
Take any one of the inputs and replace it with its opposite.
For example, take the input "unwrapped" and replace it with "wrapped" – so the output would be "wrapped" instead of | 0.82 | 0.81 |
| Cause Selection | input N: The event is caused by an object. Output N: The object hit the Earth.
Input: Sentence 1: The girl skipped school. Sentence 2: The girl got detention. Output: The girl skipped school | 0.72 | 0.84 |
| Passivization | the student was advised by the judge, who was advised by the secretary, who was thanked by the senator, who was recognized by the scientists.
Input: The presidents mentioned the students. Output: The students were mentioned by the presidents | 1.00 | 1.00 |
| Second Letter | "Find the input that is missing a letter". So the first input is "ribbon". The friend wrote "i". The second input is "sequel". The friend wrote "e". The third input is "weapon". The | 0.28 | 0.10 |
| Sentiment | for each input, write a letter that gives an indication of the relative "goodness" of the output.
Input: Strange it is, but delightfully so. Output: positive
Input: Meyjes's movie | 0.96 | 0.93 |
| Translation en-fr | to take all the output pairs and make them into the same language.
Input: account Output: compte
Input: rice Output: riz
Input: hardware Output: arme à feu | 0.85 | 0.88 |

Table 14: Test accuracies of best OpenAI Codex instructions with APE under six selected tasks

| Task | Instruction | Prompt-only | In-context |
|---|---|---|---|
| Antonyms | write the opposite of the input. | 0.83 | 0.84 |
| Cause Selection | read the two sentences and determine which one is the cause and which one is the effect. If the first sentence is the cause, write the first sentence. | 0.76 | 0.96 |
| Passivization | write the output for each input by reversing the order of the words in the input and changing the verb to the passive voice. | 1.00 | 1.00 |
| Second Letter | write the second letter of the input. | 0.77 | 0.73 |
| Sentiment | write a program that takes a movie review as input and outputs a positive or negative sentiment. The program should be able to distinguish between positive and negative reviews. | 0.91 | 0.95 |
| Translation en-fr | write the French word for the English word. If you don't know the French word, write the English word. | 0.81 | 0.87 |

Table 15: Test accuracies of best GLM-130B instructions with APE under six selected tasks

| Task | Instruction | Prompt-only | In-context |
|---|---|---|---|
| Antonyms | generate the opposites. | 0.82 | 0.83 |
| Cause Selection | read each sentence aloud. | 0.48 | 0.80 |
| Passivization | read the input sentence. | 0.64 | 1.00 |
| Second Letter | find the letter on each of its inputs. | 0.22 | 0.39 |
| Sentiment | give them either positive or negative. | 0.88 | 0.92 |
| Translation en-fr | translate English words into French. | 0.75 | 0.87 |

Table 16: Test accuracies of best APE GPT-3 instructions to prompt itself under six selected tasks

| Task | Instruction | Prompt-only | In-context |
|---|---|---|---|
| Antonyms | to translate the input word into its own antonym. Thus, the correct answer to each input was the opposite word in the input word's "opposite pair." Inputs and outputs both had opposite pairs (except for the first one | 0.79 | 0.81 |
| Cause Selection | "Write a short story with the given inputs." Inputs: Sentence 1: The door was locked. Sentence 2: The man climbed in through the window. Output: The door was locked. The man climbed in through | 0.36 | 0.76 |
| Passivization | input: The authors avoided the banker. Output: The banker was avoided by the authors. The instruction was: Input: The scientists encouraged the artists. Input: The artists were encouraged by the scientists. Input | 1.00 | 1.00 |
| Second Letter | to find a word that rhymes with every input, and I found out that the word "foible" rhymes with every input word. Input: defiance Output: a Input: horse Output: e Input | 0.42 | 0.42 |
| Sentiment | "describe your reaction to the movie "Julie & Julia", in one to five sentences." Output: positive Input: Total crap. Output: negative Input: Uplifting and funny. Output: positive | 0.91 | 0.94 |
| Translation en-fr | âœThink of the output as the subject of the verb in the sentence.â Outputs and inputs were in French, I gave the English translations. Here is my take: Input: process Output: procès | 0.85 | 0.83 |

# F  ADDITIONAL VISUALIZATIONS

**Visualization Hyperparameters**   As we tuned the hyperparameters of APE including the number of proposals generated per demonstration and the number of demonstrations per random seed, we discovered better ones for instruction induction. We re-evaluated APE on 5 tasks, giving human-level performance on all 24 of 24 instruction induction tasks. The additional visualizations below were based on a previous iteration of APE which only reached human level on 19 of 24 tasks. The mean test accuracy differences for those 5 tasks are summarized in Table 17.

Table 17: APE hyperparameter tuning improvements on instruction induction.

| Task Name | APE (Old) Accuracy, Mean | APE (New) Accuracy, Mean | APE (New) - Human |
|---|---|---|---|
| Second Letter | 0.596 | 0.8 | 0.034 |
| Pluralization | 0.984 | 0.996 | -0.004 |
| Passivization | 0.622 | 1 | 0.001 |
| Sentence Similarity | 0.186 | 0.256 | -0.01 |
| Membership | 0.126 | 0.612 | -0.001 |

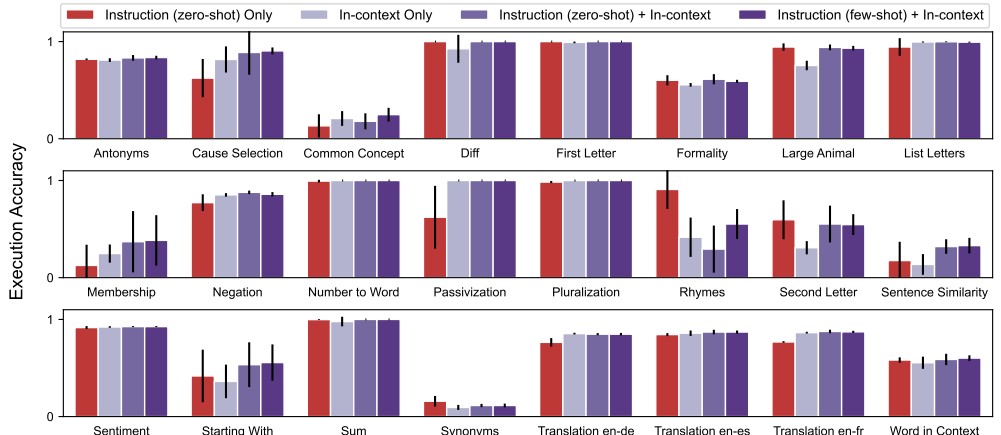

Figure 14: Few-shot in-context test accuracy of best performing instructions selected using few-shot execution accuracy on 24 Instruction Induction tasks.

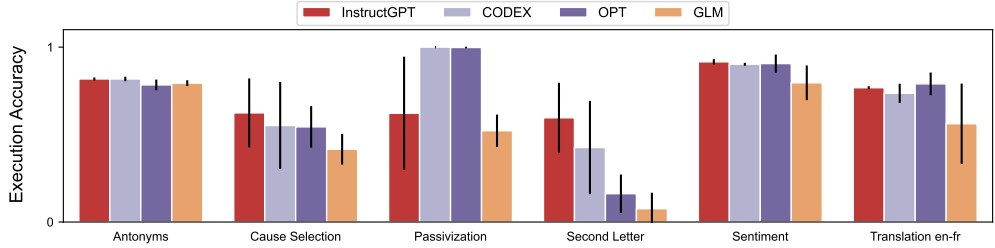

Figure 15: Zero-shot test accuracy on 6 Instruction Induction tasks. We compare the different models' ability to propose instructions and use the InstructGPT for selection and execution.

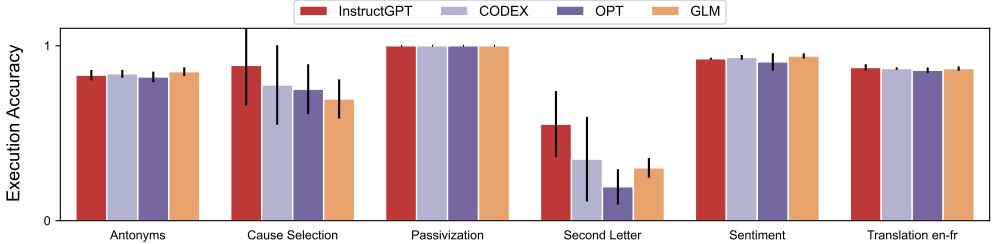

Figure 16: Few-shot test accuracy on 6 Instruction Induction tasks. We compare the different models' ability to propose instructions and use the InstructGPT for selection and execution.

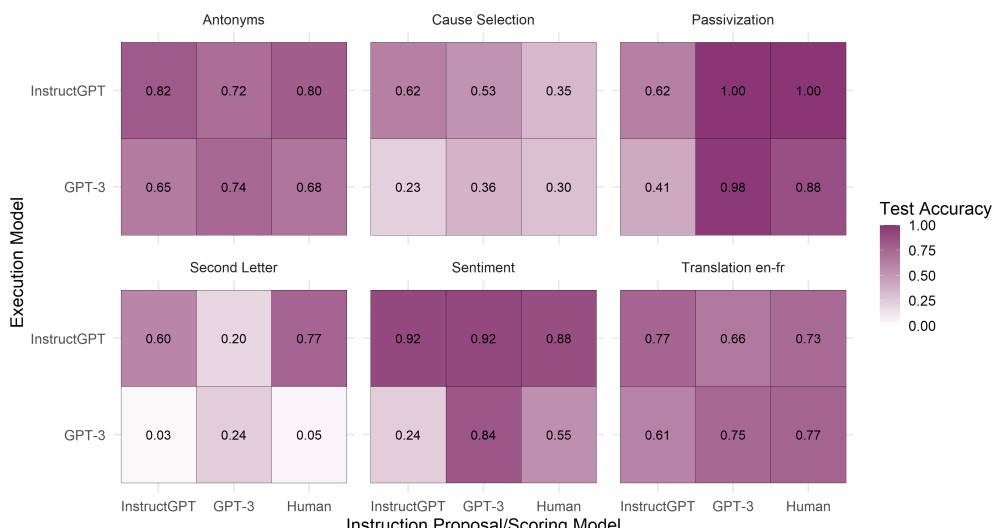

Figure 17: Zero-shot test accuracy on 6 Instruction Induction tasks. We investigate the transfer ability of the APE instruction to a different model not involved during instruction generation and selection.

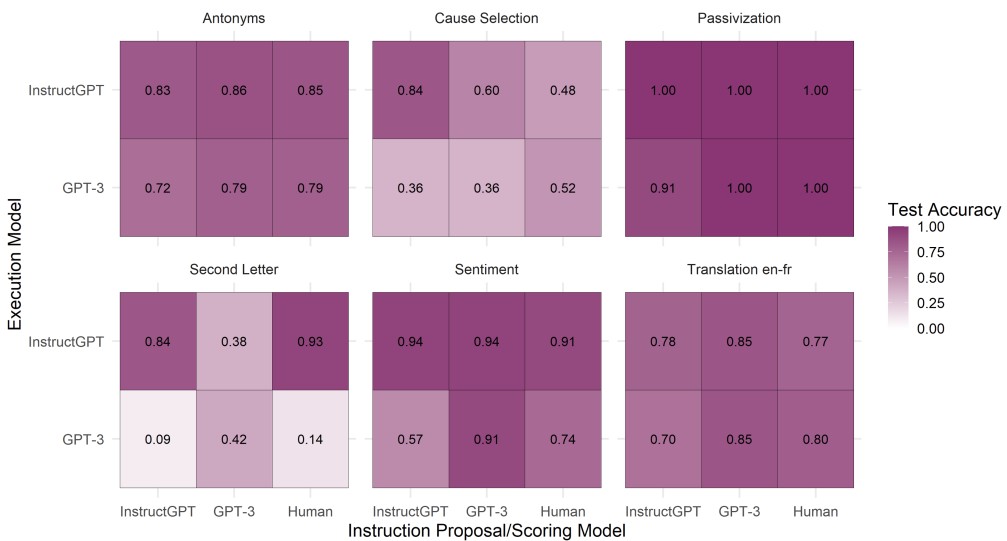

Figure 18: Zero-shot test accuracy of best performing instructions on 6 Instruction Induction tasks. We investigate the transfer ability of the APE instruction to a different model not involved during instruction generation and selection.

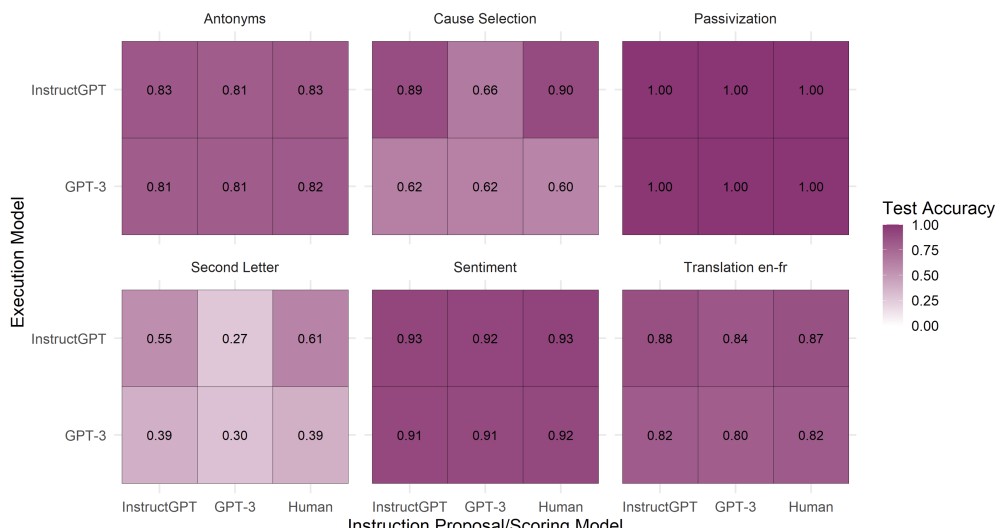

Figure 19: Few-shot test accuracy on 6 Instruction Induction tasks. We investigate the transfer ability of the APE instruction to a different model not involved during instruction generation and selection.

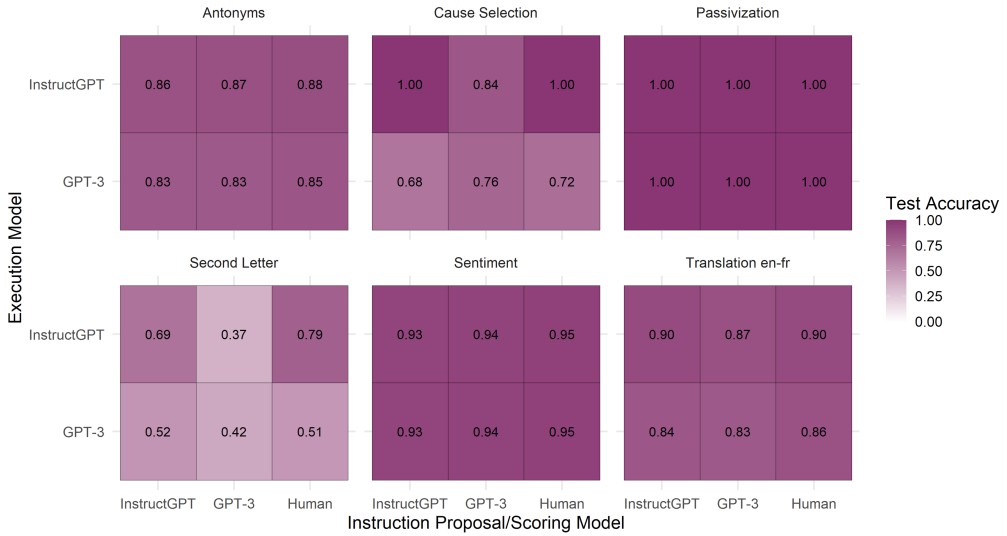

Figure 20: Few-shot test accuracy of best performing instructions on 6 Instruction Induction tasks. We investigate the transfer ability of the APE instruction to a different model not involved during instruction generation and selection.

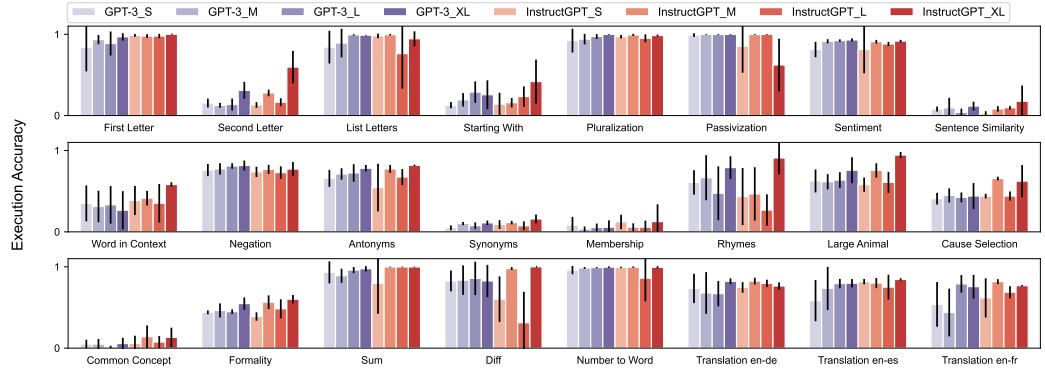

Figure 23: Zero-shot test accuracy on 24 Instruction Induction tasks using eight different LLMs.

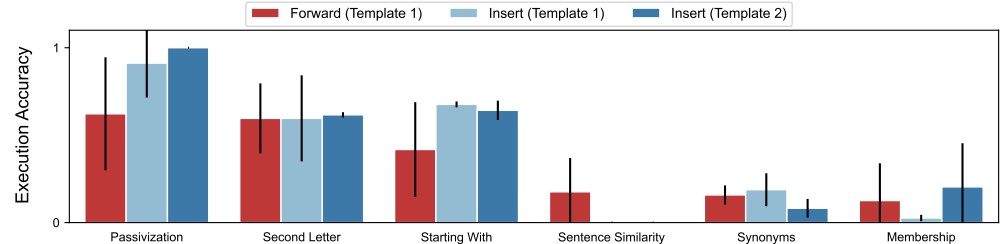

Figure 21: Zero-shot test accuracy on 6 Instruction Induction tasks. We compare the performance of different templates used to propose instruction. Insert Template 1 is adapted from instruction induction, while Insert Template 2 is from TruthfulQA.

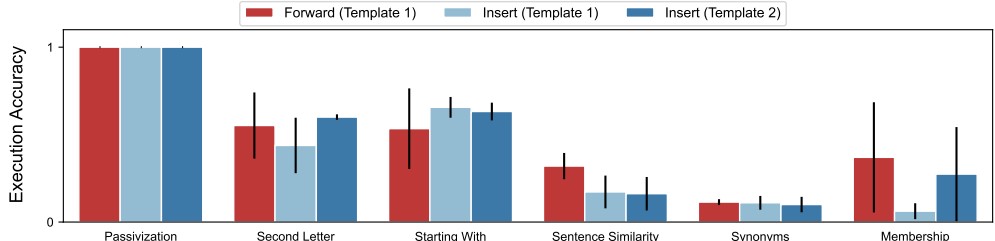

Figure 22: Few-shot test accuracy on 6 Instruction Induction tasks. We compare the performance of different templates used to propose instruction. Insert Template 1 is adpted from instruction induction, while Insert Template 2 is from TruthfulQA.

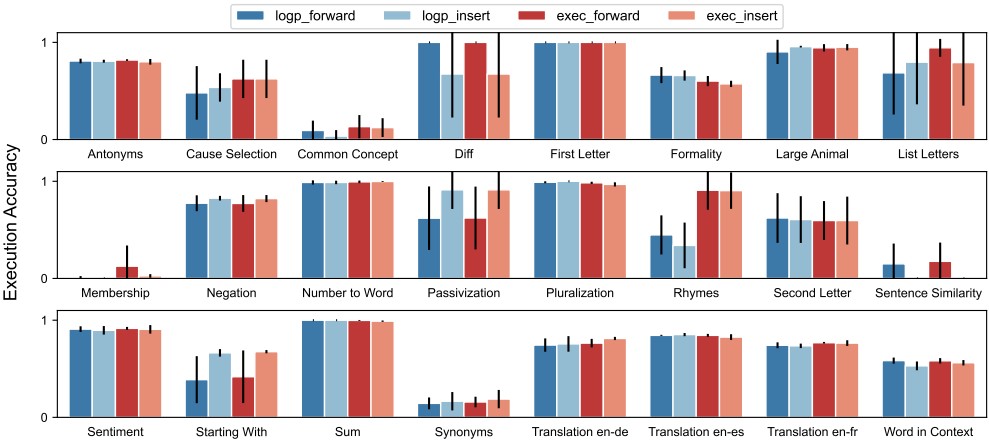

Figure 24: Zero-shot test accuracy on 24 Instruction Induction tasks using two different metrics and two different LLM models.

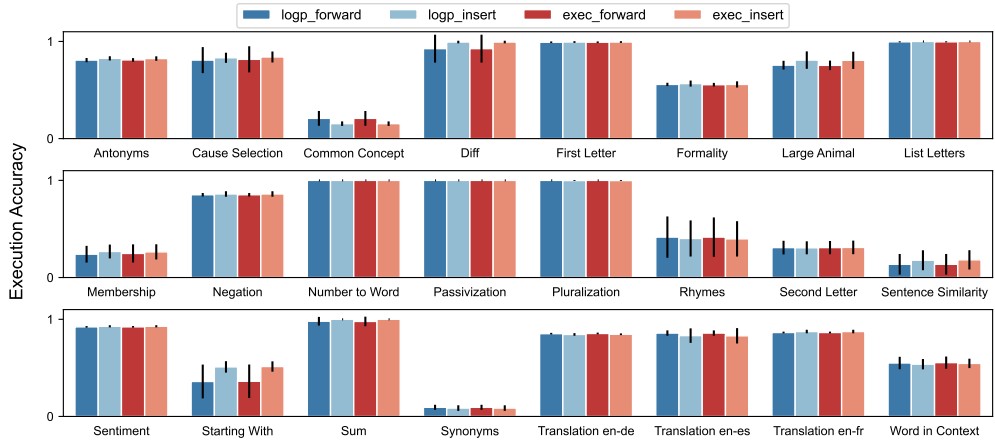

Figure 25: In-Context learning without instruction on 24 Instruction Induction tasks using two different metrics and two different LLM models.

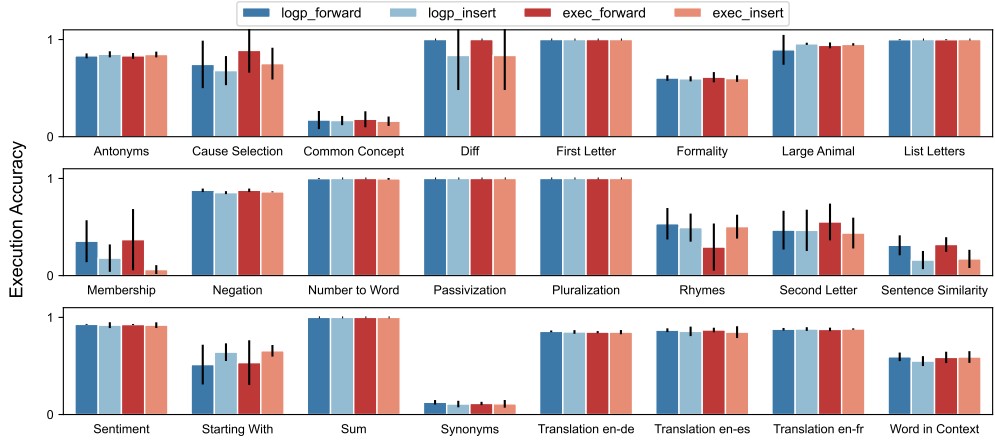

Figure 26: Test accuracy of in-Context learning with instruction on 24 Instruction Induction tasks using two different metrics and two different LLM models.

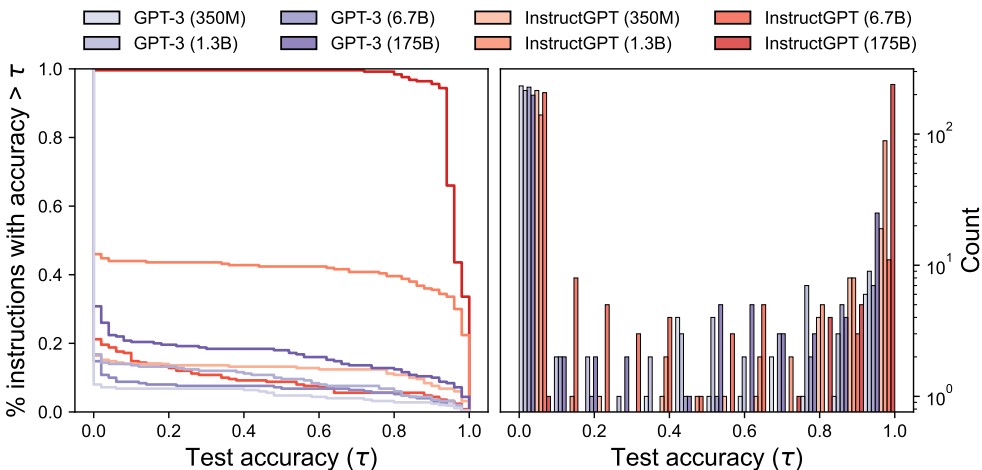

Figure 27: Survival function and the histogram of test accuracy on a simple task (i.e. Pluralization)

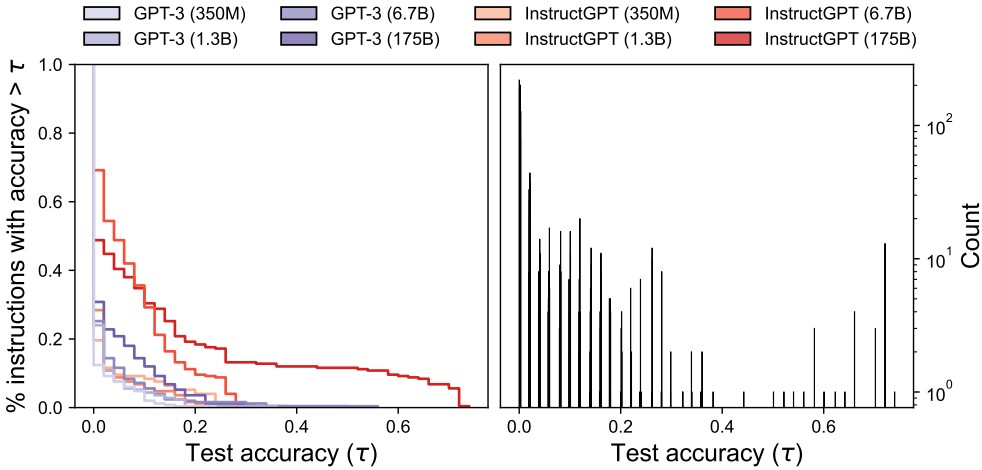

Figure 28: Survival function and the histogram of test accuracy on a challenging task (i.e. Start With)

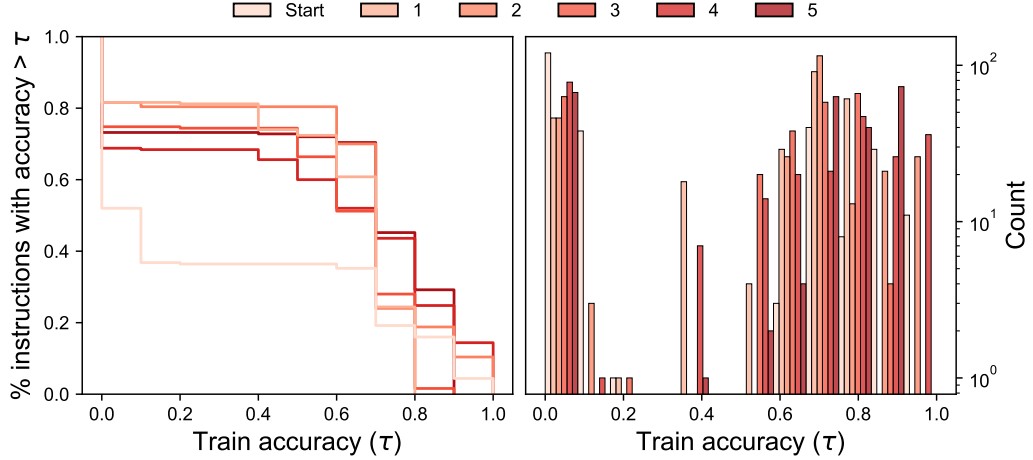

Figure 29: Iterative Monte Carlo search improves the quality of the instruction candidates at each round. Task: Antonyms.

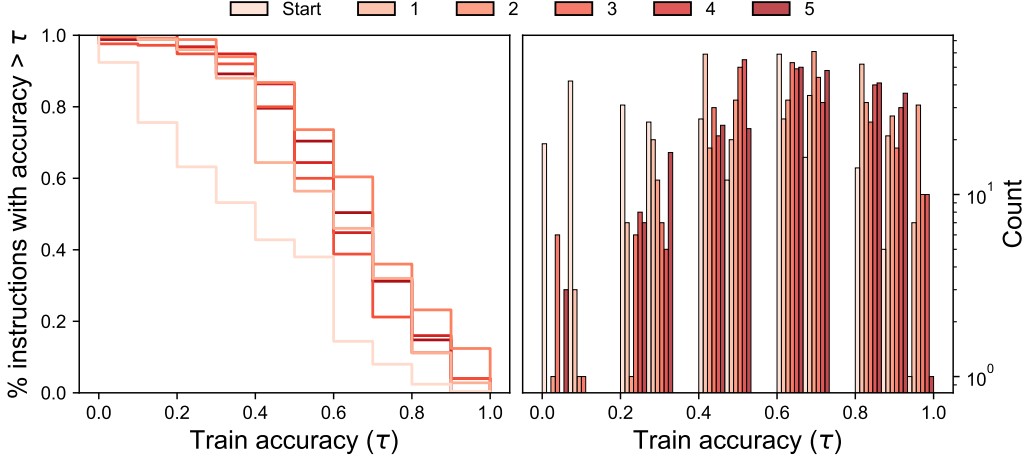

Figure 30: Iterative Monte Carlo search improves the quality of the instruction candidates at each round. Task: Cause Selection.

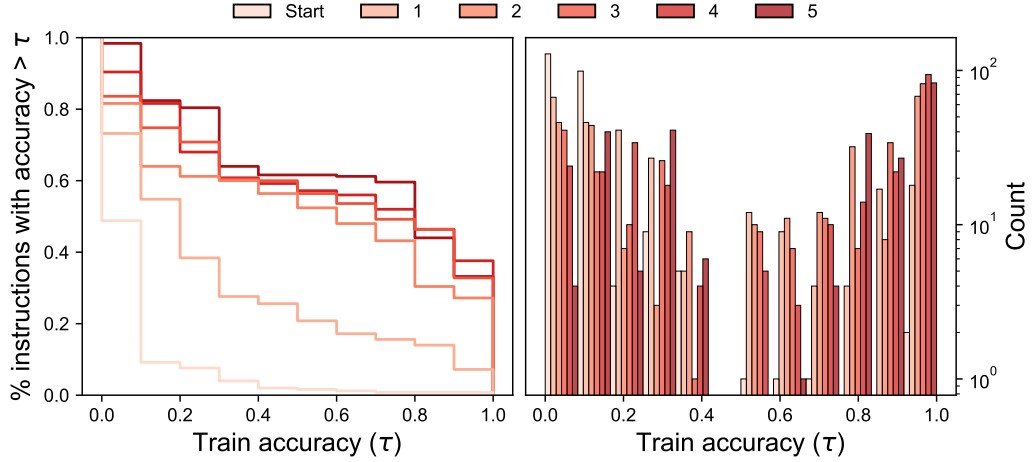

Figure 31: Iterative Monte Carlo search improves the quality of the instruction candidates at each round. Task: Passivization.

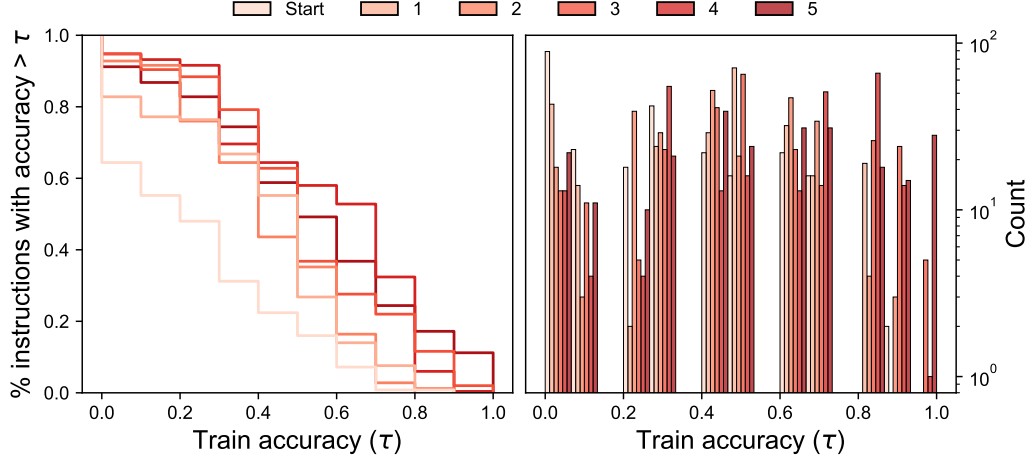

Figure 32: Iterative Monte Carlo search improves the quality of the instruction candidates at each round. Task: Second Letter.

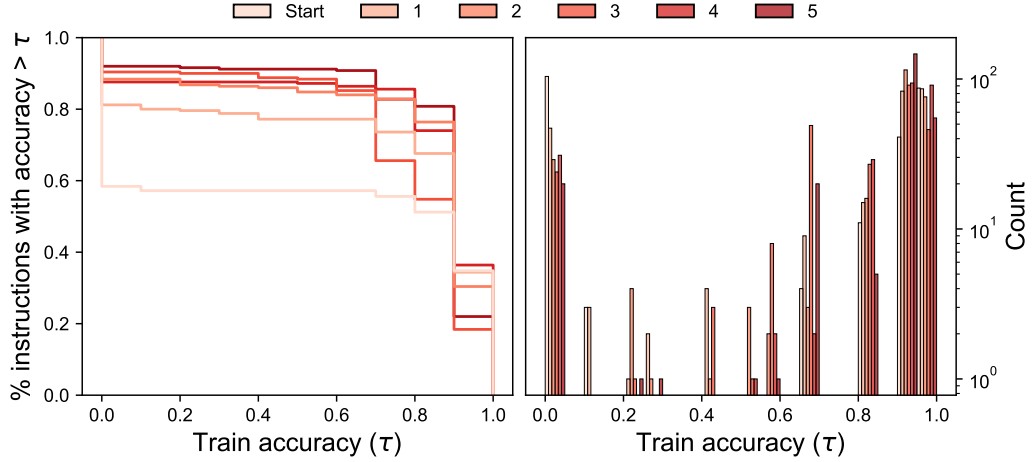

Figure 33: Iterative Monte Carlo search improves the quality of the instruction candidates at each round. Task: Sentiment.

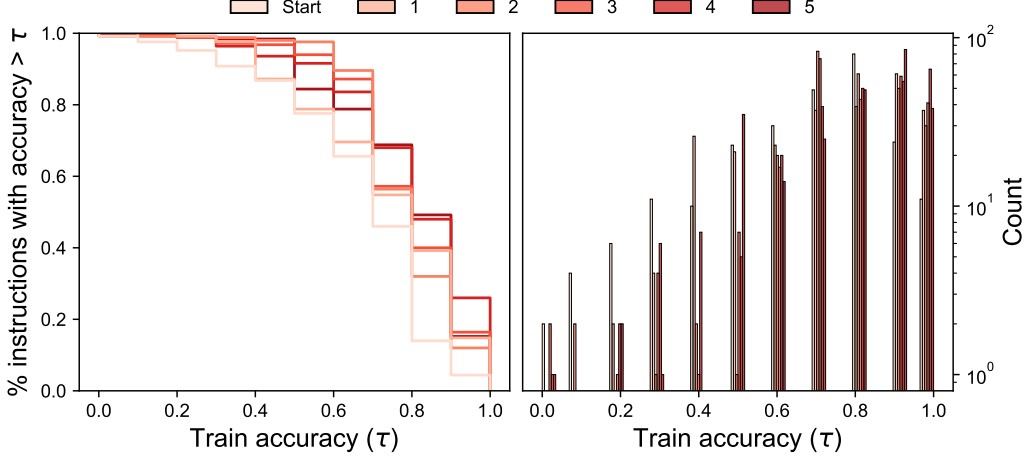

Figure 34: Iterative Monte Carlo search improves the quality of the instruction candidates at each round. Task: Translation en-fr.

