# OpenReview forum: "Large Language Models are Human-Level Prompt Engineers"
_ICLR.cc/2023/Conference — ICLR 2023 poster_

### Official Review · Reviewer_7jaz · 2022-10-23

**Confidence:** 3
**Correctness:** 3
**Technical Novelty And Significance:** 4
**Empirical Novelty And Significance:** 4
**Recommendation:** 8

**Clarity, Quality, Novelty And Reproducibility:**

The paper is not very clear, the idea is novel.
The authors do not mention to release any code, so I am not sure about reproducibility.

**Details Of Ethics Concerns:**

Same old concerns related to all large language models about toxic language gender/religious/racial biases etc.

**Strength And Weaknesses:**

The proposed idea is very interesting and this line of works trying to find the best language to query large language models is very trendy these days.
The reported results seem to be very good and for practitioners this approaches can be useful to improve performance on specific tasks.

The paper though is not very well written and there are a few parts that require some clarification.

The assumptions/intuitions in section 3.1 are not clear.
For example why "would like the model to predict the missing context before the demonstrations"?
What does it mean more versatile approach?
I understand that for every set of training pair there will be a possible different instruction, do you consolidate them?
Which one do you choose at inference time?
Real examples in the the Figures and task descriptions would simplify the reading a lot.

After the selection of instruction using different metrics, a set of final instruction are returned, how do these extraction are used? prompted all of them in an in-context learning scenario?

In section 4.1, please explain why Honovich et al. (2022) is the right benchmark for the proposed task, redirecting the reader to the original paper is not good writing, since it is not a well established benchmark.
In section 4.1 would be useful to have examples of input/output in the zero-shot and few-shot in context learning scenarios.
Why is the model only tested on the dataset released by Honovich et al. (2022), and not on more standard benchmarks like GLUE or big bench?


**Summary Of The Paper:**

The paper propose a novel way of doing prompt engineering in large language models.
The model scores a pool of instructions generated using small set of demonstrations by a large language model.
For each input output pair (x,y), the goal is to find an instruction z so that f(z,x)=y, where f is a large language model.
z is then used to prompt a large language model at inference time with and without demonstrations.
Results on two benchmarks show that the proposed approach produces instructions that are as good as instructions written by humans.
The authors also do ablation experiments to understand what is the impact of language model size.


**Summary Of The Review:**

The idea is novel and interesting.
The paper is hard to read although the approach is quite straightforward. I suggest the authors to spend some time to make it more clear.

UPDATE: The authors made the paper more clear and added a set of experiments on more benchmarks that make the paper much stronger.

---

> ### Author Response · Authors · 2022-11-18
> **Response to Reviewer 7jaz (1/2)**
>
> Thank you for your effort to review and look into every detail of our work. We appreciate that you found our paper novel. We clarify some of the confusing parts in our paper and highlight some of the new experimental results regarding your concern below. We also improve our writing and update the pipeline illustration with concrete examples. If the following comments address your concerns and the revised paper seems clearer to you, we would be grateful if you could consider increasing the review score.
>
> Here is a list changes that you may be interested in since our original submission
> - [Clarity] We update Figure 1 to include concrete examples. We add all raw templates used for model prompting to Appendix Table 5.
> - [Clarity] We clarify the motivation of reverse mode generation and compare it with forward mode template.
> - [New Experiment] We add an experiment on 21 Big-Bench instruction induction tasks to demonstrate the effectiveness of APE on more challenging tasks. (See general response or Paper Section 4.2)
> - [New Experiment] We evaluate APE on 12 zero-shot CoT reasoning tasks to show its ability to optimize non-instruction prompt so that it can be used as an assist tool for human prompt engineers. (See general response or Paper Section 4.3)
>
> > I understand that for every set of training pairs there will be a possible different instruction, do you consolidate them? Which one do you choose at inference time? Real examples in the Figures and task descriptions would simplify the reading a lot. After the selection of instructions using different metrics, a set of final instructions are returned, how do these extraction are used? prompted all of them in an in-context learning scenario?
>
> It may help to see a walk-through of how APE works, generating high-performing instruction by looking at examples. As shown in Figure 1, we perform the following steps to get a prompt with high performance.
> 1. We first split our dataset into generation, scoring, and test sets.
> 2. By sampling a few (5) examples from the generation set, we query an LLM with the following prompt in order to generate $n$ (~100) possible candidate instructions (by sampling from the LLM multiple times):
>
> ```
> I gave a friend an instruction. Based on the instruction they produced the following input-output pairs:
>
> Input: direct
> Output: indirect
>
> …
>
> Input: unpopular
> Output: popular
>
> The instruction was to <COMPLETE>
> ```
>
> 3. APE evaluates each candidate instruction on ~50 data points from the scoring set by querying the LLM in order to see how well each instruction performs on each sampled data point:
>
> ```
> Instruction: [instruction]
> Input: powerful
> Output: <COMPLETE>
> ```
> In this example, APE is using the exact-match scoring function, so in the end, each candidate instruction is scored with the fraction of data it gets exactly correct. If we instead used the likelihood scoring function, then the score would be the likelihood of the correct answers.
>
> 4. [Optional, but typically not used] Ask the LLM to generate new instruction candidates that are semantically similar to those which scored highly in the previous iteration. Repeat step 3.
>
> 5. Finally, APE returns the instruction that scored highest on the scoring split of the dataset.
>
> As shown in Equation (1), We only select one prompt with the highest score for each task and the same prompt is used at inference time for all testing data points. We adopt “Instruction: [APE Prompt]\n\nQ: [Test Input]\nA:<Model complete>” for zero-shot evaluation and “Instruction: [APE Prompt][Few-shot Examples]\n\nQ: [Test Input]\nA:<Model complete>” for few-shot evaluation. We hope this can clarify some of your questions regarding the process.

---

> > ### Author Response · Authors · 2022-11-18
> > **Response to Reviewer 7jaz (2/2)**
> >
> > > The assumptions/intuitions in section 3.1 are not clear. For example why "would like the model to predict the missing context before the demonstrations"? What does it mean more versatile approach?
> >
> > The “missing context” here refers to the instruction prompt, which is what we want to infer. It is common in large language model prompting that an instruction is first given before showing any concrete examples. If we want to adopt this form without any modification, then the model needs to predict the missing context before the demonstrations. We call this type of instruction inference reverse mode generation. In contrast, if the model only has left-to-right generation ability, then we have to reformat the demonstration cleverly so that the predicted instruction is at the end of the prompt for the LLMs to fill. Thus, we name this type of instruction generation forward mode generation. However, this reformatting is not easy and requires some human effort, so it is not as flexible as the reverse mode generation.
> >
> > > In section 4.1, please explain why Honovich et al. (2022) is the right benchmark for the proposed task, redirecting the reader to the original paper is not good writing, since it is not a well established benchmark. In section 4.1 would be useful to have examples of input/output in the zero-shot and few-shot in context learning scenarios.
> >
> > The instruction induction task was first proposed by Honovich et al. (2022), which includes 24 NLP tasks with verified prompts written by human prompt engineers. We choose Honovich et al. (2022) because it proposes an appropriate "greedy" method to tackle these tasks, which can serve as a baseline comparison to APE. To further demonstrate the effectiveness of our proposed method, we have included the experiments on BigBench and zero-shot CoT reasoning tasks (See general response).  Since the goal of our paper is to propose a novel method for automatic prompt engineering rather than propose a new dataset or benchmark, we do not elaborate on these tasks in the main text. However, we recognize that added clarity about our benchmark tasks is important and therefore we have added a detailed description of each task in Appendix B. Examples of input/output pairs for zero-shot and few-shot learning can be found in Appendix B.
> >
> > > Why is the model only tested on the dataset released by Honovich et al. (2022), and not on more standard benchmarks like GLUE or big bench?
> >
> > As discussed above, the dataset in Honovich et al. (2022) allows us to easily compare the quality of prompts generated with APE to those made by humans. To further address your concern, we have added a set of new experiments to show that APE performs well on a diverse and challenging set of additional tasks: a set of 21 BigBench tasks (curated so that they include human-written prompts for comparison) and 12 zero-shot-CoT reasoning datasets (See general response). In these new tasks, APE achieves comparable or better performance to the human provided prompts, even beating extensively-tuned human-prompts on CoT-zero-shot reasoning tasks.
> >
> > > The authors do not mention to release any code, so I am not sure about reproducibility.
> >
> > You can find our code, a simple jupyter notebook, and simple UI demo in the supplementary Material. This code will be released open-source upon publication to allow anyone to experiment with automated prompt engineering.
> >
> > > Same old concerns related to all large language models about toxic language gender/religious/racial biases etc.
> >
> > We recognize that large language models can be biased and propose safety challenges in the above areas and more. However, as the purpose of our work is to enable users to steer the outputs of LLMs using prompts (which are human-interpretable), we believe that our work is a step towards improvement in these areas. For example, in TruthfulQA, we show that APE can be used to find prompts that steer the model to behave more truthfully and informative. The same technique can be potentially applied to reduce the toxic language, gender/religious/racial biases etc.

---

### Official Review · Reviewer_3MTv · 2022-10-24

**Confidence:** 4
**Clarity, Quality, Novelty And Reproducibility:** Please refer to Strengths.
**Correctness:** 3
**Technical Novelty And Significance:** 3
**Empirical Novelty And Significance:** 3
**Recommendation:** 6

**Strength And Weaknesses:**

Strength:
- The paper is well-written and provide a lot of insights that can help other researchers who are working on similar area.
- The selection + filtering is a new addition to prior work done by [1] which improves the results significantly.
- Experiments are well-done and multiple ablations are performed to show effectiveness of various components of their approach.
-The idea is similar to automatic prompt tuning methods such as prefix-tuning, however, the resulting instructions are in natural language which is more interpretable than other approaches which generate embeddings. Also this approach doesn't require fine-tuning which is a plus.

Weakness:
- I think the tasks studied in this paper could be more challenging. Taking a look at Table1, most tasks require surface-level understanding of language syntax. Translation task is only done for single words. Semantic tasks from GLUE are at 90+% accuracy with simple transformer models. Few tasks here require complex reasoning, or deeper language understanding. I think those tasks require creating more nuanced instructions and seeing how this approach compares against humans would be very interesting. Until then. it's unclear to me how this approach generalizes beyond simple tasks.
- To build on my previous point, looking at figure 3, results on relatively harder tasks such as sentence similarity and membership show APE is doing worse than humans and vanilla generated prompts. The authors acknowledge this in the paper: "Additionally, tasks such as Membership and Second Letter are intrinsically challenging for the model, and APE consistently performs worse than humans." I think the paper can significantly benefit from further studies on tasks such as summarization, intent classification and slot filling, etc. that are more challenging.

[1] Or Honovich, Uri Shaham, Samuel R Bowman, and Omer Levy. Instruction induction: From few
examples to natural language task descriptions. arXiv preprint arXiv:2205.10782, 2022.

**Summary Of The Paper:**

This paper proposes using LLMs to generate instructions that will be used as prompts for solving tasks with LLMs.
Authors use a filtering + selection pipeline to choose top candidates according to various metrics. They compare their method in zero and few shot settings with 1) human-engineered prompts 2) LLM generated outputs without the filtering and section process.
They evaluate their approach on several NLP datasets and show it achieves similar results to 1) and improves over 2).
They do a comprehensive analysis of their experiments that shed light on important components of their approach.


**Summary Of The Review:**

Given the Strengths and Weaknesses mentioned above, I think this is a valuable paper and authors have done a lot of work to ensure the accuracy of their claims. However, I'm not quite convinced if this approach can scale to more challenging NLP tasks. I hope authors address this issue in the updated version of the paper. I can revisit my score given the new results.

---

> ### Author Response · Authors · 2022-11-18
> **Response to Reviewer 3MTv**
>
> Thank you for your effort to review and look into every detail of our work. We appreciate that you found our paper well-written and insightful and recommend accepting our paper. If the following comments address your concerns, we would be grateful if you could consider increasing the review score.
>
> > I think the tasks studied in this paper could be more challenging. Taking a look at Table1, most tasks require surface-level understanding of language syntax. Translation task is only done for single words. Semantic tasks from GLUE are at 90+% accuracy with simple transformer models. Few tasks here require complex reasoning, or deeper language understanding. I think those tasks require creating more nuanced instructions and seeing how this approach compares against humans would be very interesting.
>
> > …
>
> > I think the paper can significantly benefit from further studies on tasks such as summarization, intent classification and slot filling, etc. that are more challenging.
>
> To address your concern regarding challenging tasks, we have added a set of new experiments where we evaluate APE on 21 BigBench tasks and 12 zero-shot-CoT reasoning datasets (See general response). In these new tasks, APE achieves comparable or better performance to the human provided prompts, even beating the tuned human-prompts on CoT-zero-shot reasoning tasks. We highlight some of the results below.
>
> - On the zero-shot-CoT reasoning experiment (e.g., arithmetic, commonsense, symbolic, and other logical reasoning tasks), APE is able to find a prompt that far-outperforms “Let’s think step by step” (which remained the best-performing prompt for ~6 months) on MultiArith and GSM8K with “Let’s work this out in a step by step way to be sure we have the right answer.”
> - In terms of the 21 BigBench tasks including various reasoning (e.g., arithmetic, commonsense, symbolic, and other logical reasoning tasks), emotional understanding, context-free question answering, reading comprehension, summarization, and algorithms tasks, APE achieves equal or better performance than the human prompt baseline on 17/21 tasks.
>
> > To build on my previous point, looking at figure 3, results on relatively harder tasks such as sentence similarity and membership show APE is doing worse than humans and vanilla generated prompts. The authors acknowledge this in the paper: "Additionally, tasks such as Membership and Second Letter are intrinsically challenging for the model, and APE consistently performs worse than humans."
>
> In running further experiments with APE, we found significantly better hyperparameters (higher number of candidate instructions generated using several different sets of examples). We re-evaluated APE with these better hyperparameters on the 5 instruction induction tasks where APE underperformed the human baseline in the original submission. We now find that APE can achieve human level performance on all of our original tasks. APE achieves a solidly super-human performance on the instruction induction task with an IQM of 0.810 vs humans’ 0.749.
> |      Task Name      | Mean Acc (old) | Mean Acc (new) | APE-Human |
> |:-------------------:|:--------------:|:-------------:|:---------:|
> |    Second Letter    |      0.596     |      0.8      |   0.034   |
> |    Pluralization    |      0.984     |     0.996     |   -0.004  |
> |    Passivization    |      0.622     |      1.0      |   0.001   |
> | Sentence Similarity |      0.186     |     0.256     |   -0.01   |
> |      Membership     |      0.126     |     0.612     |   -0.001  |

---

> ### Author Response · Authors · 2022-12-08
> **Reminder (1/2)**
>
> Dear Reviewer 3MTv,
>
> We appreciate your careful review and the valuable feedback that you provided for our paper. We have taken your comments into consideration and have made several revisions to our work in order to address the issues that you raised. In light of these improvements, we kindly request that if your concerns were addressed, you consider raising your score. We also want to remind you that the discussion phase will end soon, so we would greatly appreciate your support in this regard before the deadline.
>
> To help you understand the changes we have made, we have provided a summary of three main improvements below. If you have any additional questions or concerns, please do not hesitate to contact us. We are committed to working with you to ensure that our paper meets the high standards of the conference.
>
> > To address concerns regarding our empirical results, we expanded our evaluation to include two more benchmark suites: a newly curated BigBench Instruction Induction subset (21 tasks) and the Zero-shot Chain of Thought tasks (12 tasks). We also re-ran our original Instruction Induction experiment with updated hyperparameters and achieve super-human performance on 24 out of 24 tasks. (See more details in General Response)
>
> 1) BigBench Instruction Induction subset (new tasks): We propose and curate BigBench Instruction Induction (BBII): a clean, and tractable subset of 21 tasks that satisfy the instruction induction format. We find that APE retains its strong performance on this new benchmark suite. When compared to human prompts, APE generated prompts improve or match zero-shot performance on 17 out of 21 tasks. We report the normalized preferred metric defined in [1]. Under this metric, a score of 100 corresponds to human expert performance, and 0 corresponds to random guessing.
>
> | Task                              | Human    | APE      |
> |-----------------------------------|:----------:|:----------:|
> | causal judgment                   | 18.0          | 18.0     |
> | disambiguation qa                 | -0.4          | **5.6**  |
> | dyck languages                    | 3.0           | **18.0** |
> | epistemic reasoning               | 36.0          | **38.0** |
> | gender inclusive sentences german | 13.0          | **22.0** |
> | implicatures                      | 60.0          | 60.0     |
> | linguistics puzzles               | 0.0           | 0.0      |
> | logical fallacy detection         | **24.0**      | 12.0     |
> | movie recommendation              | -2.7          | **12.0** |
> | navigate                          | -8.0          | **12.0** |
> | object counting                   | 2.0           | **44.0** |
> | operators                         | **48.0**      | 47.0     |
> | presuppositions as nli            | **13.0**      | 5.5      |
> | question selection                | -2.6          | **-0.9** |
> | ruin names                        | **1.3**       | -14.7    |
> | snarks                            | 2.0           | **4.0**  |
> | sports understanding              | 36.0          | 36.0     |
> | tense                             | 84.0          | **85.0** |
> | winowhy                           | -12.0         | **12.0** |
> | word sorting                      | 11.0          | **30.0** |
> | word unscrambling                 | 10.0          | **15.0** |

---

> > ### Author Response · Authors · 2022-12-08
> > **Reminder (2/2)**
> >
> > 2) We follow [2] to evaluate APE on 12 reasoning tasks from four categories of reasoning tasks: arithmetic, commonsense, symbolic, and other logical reasoning tasks. As shown in the table below. APE is able to find a prompt “Let’s work this out in a step by step way to be sure we have the right answer” that ranks first among the hand crafted prompts in Table 4 of [2], where the previous SOTA (which remained the best known CoT-triggering prompt for ~6 months), “Let’s think step by step”, is the most famous recent example of prompt engineering. The APE-generated prompt improves text-davinci-002’s Zero-Shot-CoT performance on MultiArith from 78.7 to 82.0 and improves GSM8K from 40.7 to 43.0.
> >
> > | No. | Category       | Zero-shot CoT Trigger Prompt                                                   | Accuracy |
> > |:-----:|:----------------:|--------------------------------------------------------------------------------|:----------:|
> > | 1   | APE            | Let's work this out in a step by step way to be sure we have the right answer. | **82.0** |
> > | 2   | Human-Designed | Let's think step by step. (*1)                                                 | 78.7          |
> > | 3   |                | First, (*2)                                                                    | 77.3          |
> > | 4   |                | Let's think about this logically.                                              | 74.5          |
> > | 5   |                | Let's solve this problem by splitting it into steps. (*3)                      | 72.2          |
> > | 6   |                | Let's be realistic and think step by step.                                     | 70.8          |
> > | 7   |                | Let's think like a detective step by step.                                     | 70.3          |
> > | 8   |                | Let's think                                                                    | 57.5          |
> > | 9   |                | Before we dive into the answer,                                                | 55.7          |
> > | 10  |                | The answer is after the proof.                                                 | 45.7          |
> > | -   |                | (Zero-shot)                                                                    | 17.7          |
> >
> > 3) Instruction Induction (original tasks): APE achieves a solidly super-human performance on the instruction induction task with an IQM of 0.810 vs humans’ 0.749. Below, we show the new results on the five tasks where APE previously underperformed humans.
> >
> > |      Task Name      | Mean Acc (old) | Mean Acc (new) | APE-Human |
> > |:-------------------:|:--------------:|:-------------:|:---------:|
> > |    Second Letter    |      0.596     |      0.8      |   0.034   |
> > |    Pluralization    |      0.984     |     0.996     |   -0.004  |
> > |    Passivization    |      0.622     |      1.0      |   0.001   |
> > | Sentence Similarity |      0.186     |     0.256     |   -0.01   |
> > |      Membership     |      0.126     |     0.612     |   -0.001  |
> >
> >
> > [1] Aarohi Srivastava, Abhinav Rastogi, Abhishek Rao, Abu Awal Md Shoeb, Abubakar Abid, Adam Fisch, Adam R Brown, Adam Santoro, Aditya Gupta, Adrià Garriga-Alonso, et al. Beyond the imitation game: Quantifying and extrapolating the capabilities of language models. arXiv preprint arXiv:2206.04615, 2022.
> >
> > [2] Takeshi Kojima, Shixiang Shane Gu, Machel Reid, Yutaka Matsuo, and Yusuke Iwasawa. Large language models are zero-shot reasoners. arXiv preprint arXiv:2205.11916, 2022.

---

### Official Review · Reviewer_y9rU · 2022-10-25

**Confidence:** 4
**Correctness:** 3
**Technical Novelty And Significance:** 3
**Empirical Novelty And Significance:** 3
**Recommendation:** 6

**Clarity, Quality, Novelty And Reproducibility:**

The paper is easy to follow, the writing is clear as well as the plots. The method and the experiments are sound, I appreciated the multiple runs to estimate the confidence of the accuracies. The exaggerated hype is a smudge in a research work of good quality.
The broad topic of prompt engineering is very novel, while the specific idea of an automatic prompt engineer was first explored in the very recent Hononovic et al. (2022). I still consider the novelty of this work satisfactory, given the new method presented and the general novelty of the area of research.


**Strength And Weaknesses:**

Strengths
- The value of a good automatic prompt engineer is so evident that needs no explanation, this paper is definitely of interest to the community.
- The method is sound, and the discussion and results are valuable.
- The related works section does a good job in connecting a very new work built on top of LLMs to established research fields such as Program Synthesis.
- The paper is clear and well written, it is easy to understand the problem, the method, and the experiments.
Weaknesses
- The paper does not achieve the expectations set by the title and abstract. I was expecting challenging tasks where humans struggle to find a proper prompt (like leading Dall-e to generate a horse riding an astronaut, which was cited as a limitation of Dall-e until someone found the proper prompt) but the proposed method can automatically find it. Instead, most of the tasks can be solved with human-level accuracy using just the enumeration of a few input/output pairs as prompts (Fig. 4). These pairs would be available in any case since they are needed by the proposed method.


**Summary Of The Paper:**

The authors propose a method to generate prompts that solve textual tasks defined by Input/Output pairs through a Large Language Model.
This paradigm was introduced by Hononovic et al. (2022), as well as the 24 tasks used for evaluation. The main element of novelty here is the algorithm based on search and validation of the best prompt among a set of generated candidates, rather than the naive greedy selection used in Hononovic et al. (2022).


**Summary Of The Review:**

My recommendation is to accept this paper, the research is of interest and well performed.
Still, I would recommend changing the title to be more descriptive and less pompous.
Right now it suggests that finding the appropriate prompt to solve a task is no longer a problem since LLMs can do that to the same level as humans.I would say that we are not still there (e.g. https://twitter.com/goodside/status/1581805503897735168).

This work made me think of https://arxiv.org/abs/2201.10222, the *natural language program synthesis* problem reminded me the *Explanatory Learning* problem, while the APE is similar to the CRN approach there, maybe you could find this paper interesting.

---

> ### Author Response · Authors · 2022-11-18
> **Response to Reviewer y9rU (1/2)**
>
> Thank you for taking the time to review and look into every detail of our work. We appreciate that you found our paper well-written and novel and recommend accepting our paper. If the following comments address your concerns, we would be grateful if you could consider increasing the review score.
>
> > The paper does not achieve the expectations set by the title and abstract. I was expecting challenging tasks where humans struggle to find a proper prompt (like leading Dall-e to generate a horse riding an astronaut, which was cited as a limitation of Dall-e until someone found the proper prompt) but the proposed method can automatically find it.
>
> > …
>
> > I would recommend changing the title to be more descriptive and less pompous. Right now it suggests that finding the appropriate prompt to solve a task is no longer a problem since LLMs can do that to the same level as humans.I would say that we are not still there (e.g. https://twitter.com/goodside/status/1581805503897735168).
>
> Although we agree with your position that APE is not human-level in all cases, and that prompt engineering is not a “solved” problem after our paper, we respectfully disagree that the title is inappropriate, for the following reasons:
> - The term prompt-engineering has many interpretations, so we have updated our paper to specify our exact scope. While multiple prompts can be chained together and LLMs can be given access to various tools (such as in the Twitter thread you linked), prompt engineering as discussed in our paper relates only to optimizing the language in a single prompt in order to elicit the best possible performance. We have also included discussion of how APE can be used as a tool by human prompt engineers in order to optimize individual parts of such pipelines, such as in our zero-shot CoT experiments (Section 4.3).
> - With our existing and newly added experiments, we are confident that APE performs this task at or above human-level. This is for two reasons. First, we directly compare the performance of human prompt engineers with APE empirically across 24(+21) instruction induction tasks and find that APE outperforms humans on a vast majority of these tasks. Second, although this is certainly subjective, we found that several of the prompts generated by APE use prompting methods that we otherwise would not have thought of. For example, for our basic translation task, APE tells GPT to “translate using the Google Translate API,” improving performance significantly.
> - To address your criticism regarding challenging tasks, we have added a set of new experiments, where we evaluate APE on 21 Big-Bench tasks and 12 reasoning tasks (See general response). In these new tasks, APE achieves comparable or better performance to the human provided prompts, even beating an extensively-tuned human-prompt (“Let’s think step by step”) on CoT-zero-shot reasoning tasks.
>
> For the above reasons, we are reluctant to change the title. However, should there be reviewer consensus that the title is misleading/inappropriate, we will consider alternatives.

---

> > ### Author Response · Authors · 2022-11-18
> > **Response to Reviewer y9rU (2/2)**
> >
> > > Most of the tasks can be solved with human-level accuracy using just the enumeration of a few input/output pairs as prompts. These pairs would be available in any case since they are needed by the proposed method.
> >
> > We recognize that this will likely be a common question when reading our work, so we have added more discussion about it in Section 4. The purpose of APE is not to replace or out-perform few-shot learning (although APE prompts happen to outperform few-shot learning on several tasks). Instead, we aim to find optimal prompts within a specific token budget that perform well. APE provides value since: 1) it often finds clever prompts that can teach us more about how to get the most out of LLMs; and 2) APE can find prompts that perform well while reducing tokens and dramatically minimizing costs compared to few-shot learning.
> >
> > **Insights into Prompt Engineering**. APE often finds prompts that surprise us and give us new insight into prompt engineering. For example, on our English to Spanish translation task, APE generates prompts such as “Translate the word into Spanish using the Google Translate API,” which outperforms the human prompt “Translate the English words into Spanish.” It turns out, mentioning Google Translate improves InstructGPT’s ability to perform translation. In some preliminary experiments, this insight translates to tougher translation benchmarks such as WMT14-en-de, where mentioning Google Translate gives a slight increase of 0.17 BLEU score. This is also the case in our new zero-shot-CoT reasoning experiment, where APE finds a prompt that far-outperforms “Let’s think step by step” (which remained the best-performing prompt for ~6 months) on MultiArith and GSM8K with “Let’s work this out in a step by step way to be sure we have the right answer.”
> >
> > **Reduced Cost**. As shown in Appendix D, the instructions found by APE from InstructGPT are much more token efficient compared to in-context examples. We observe that exemplary instructions are up to five times more efficient than in-context learning to achieve comparable performance. Therefore, though the process to find the APE instruction introduces some additional costs, using APE achieves a much lower amortized cost when we need to evaluate a large number of queries at inference time. We can view the APE instruction as a highly condensed prompt distilled from the few-shot in-context demonstrations.

---

> ### Author Response · Authors · 2022-12-08
> **Reminder (1/2)**
>
> Dear Reviewer y9rU,
>
> We appreciate your careful review and the valuable feedback that you provided for our paper. We have taken your comments into consideration and have made several revisions to our work in order to address the issues that you raised. In light of these improvements, we kindly request that if your concerns were addressed, you consider raising your score. We also want to remind you that the discussion phase will end soon, so we would greatly appreciate your support in this regard before the deadline.
>
> To help you understand the changes we have made, we have provided a summary of three main improvements below. If you have any additional questions or concerns, please do not hesitate to contact us. We are committed to working with you to ensure that our paper meets the high standards of the conference.
>
> > To address concerns regarding our empirical results, we expanded our evaluation to include two more benchmark suites: a newly curated BigBench Instruction Induction subset (21 tasks) and the Zero-shot Chain of Thought tasks (12 tasks). We also re-ran our original Instruction Induction experiment with updated hyperparameters and achieve super-human performance on 24 out of 24 tasks. (See more details in General Response)
>
> 1) BigBench Instruction Induction subset (new tasks): We propose and curate BigBench Instruction Induction (BBII): a clean, and tractable subset of 21 tasks that satisfy the instruction induction format. We find that APE retains its strong performance on this new benchmark suite. When compared to human prompts, APE generated prompts improve or match zero-shot performance on 17 out of 21 tasks. We report the normalized preferred metric defined in [1]. Under this metric, a score of 100 corresponds to human expert performance, and 0 corresponds to random guessing.
>
> | Task                              | Human    | APE      |
> |-----------------------------------|:----------:|:----------:|
> | causal judgment                   | 18.0          | 18.0     |
> | disambiguation qa                 | -0.4          | **5.6**  |
> | dyck languages                    | 3.0           | **18.0** |
> | epistemic reasoning               | 36.0          | **38.0** |
> | gender inclusive sentences german | 13.0          | **22.0** |
> | implicatures                      | 60.0          | 60.0     |
> | linguistics puzzles               | 0.0           | 0.0      |
> | logical fallacy detection         | **24.0**      | 12.0     |
> | movie recommendation              | -2.7          | **12.0** |
> | navigate                          | -8.0          | **12.0** |
> | object counting                   | 2.0           | **44.0** |
> | operators                         | **48.0**      | 47.0     |
> | presuppositions as nli            | **13.0**      | 5.5      |
> | question selection                | -2.6          | **-0.9** |
> | ruin names                        | **1.3**       | -14.7    |
> | snarks                            | 2.0           | **4.0**  |
> | sports understanding              | 36.0          | 36.0     |
> | tense                             | 84.0          | **85.0** |
> | winowhy                           | -12.0         | **12.0** |
> | word sorting                      | 11.0          | **30.0** |
> | word unscrambling                 | 10.0          | **15.0** |

---

> > ### Author Response · Authors · 2022-12-08
> > **Reminder (2/2)**
> >
> > 2) We follow [2] to evaluate APE on 12 reasoning tasks from four categories of reasoning tasks: arithmetic, commonsense, symbolic, and other logical reasoning tasks. As shown in the table below. APE is able to find a prompt “Let’s work this out in a step by step way to be sure we have the right answer” that ranks first among the hand crafted prompts in Table 4 of [2], where the previous SOTA (which remained the best known CoT-triggering prompt for ~6 months), “Let’s think step by step”, is the most famous recent example of prompt engineering. The APE-generated prompt improves text-davinci-002’s Zero-Shot-CoT performance on MultiArith from 78.7 to 82.0 and improves GSM8K from 40.7 to 43.0.
> >
> > | No. | Category       | Zero-shot CoT Trigger Prompt                                                   | Accuracy |
> > |:-----:|:----------------:|--------------------------------------------------------------------------------|:----------:|
> > | 1   | APE            | Let's work this out in a step by step way to be sure we have the right answer. | **82.0** |
> > | 2   | Human-Designed | Let's think step by step. (*1)                                                 | 78.7          |
> > | 3   |                | First, (*2)                                                                    | 77.3          |
> > | 4   |                | Let's think about this logically.                                              | 74.5          |
> > | 5   |                | Let's solve this problem by splitting it into steps. (*3)                      | 72.2          |
> > | 6   |                | Let's be realistic and think step by step.                                     | 70.8          |
> > | 7   |                | Let's think like a detective step by step.                                     | 70.3          |
> > | 8   |                | Let's think                                                                    | 57.5          |
> > | 9   |                | Before we dive into the answer,                                                | 55.7          |
> > | 10  |                | The answer is after the proof.                                                 | 45.7          |
> > | -   |                | (Zero-shot)                                                                    | 17.7          |
> >
> > 3) Instruction Induction (original tasks): APE achieves a solidly super-human performance on the instruction induction task with an IQM of 0.810 vs humans’ 0.749. Below, we show the new results on the five tasks where APE previously underperformed humans.
> >
> > |      Task Name      | Mean Acc (old) | Mean Acc (new) | APE-Human |
> > |:-------------------:|:--------------:|:-------------:|:---------:|
> > |    Second Letter    |      0.596     |      0.8      |   0.034   |
> > |    Pluralization    |      0.984     |     0.996     |   -0.004  |
> > |    Passivization    |      0.622     |      1.0      |   0.001   |
> > | Sentence Similarity |      0.186     |     0.256     |   -0.01   |
> > |      Membership     |      0.126     |     0.612     |   -0.001  |
> >
> >
> > [1] Aarohi Srivastava, Abhinav Rastogi, Abhishek Rao, Abu Awal Md Shoeb, Abubakar Abid, Adam Fisch, Adam R Brown, Adam Santoro, Aditya Gupta, Adrià Garriga-Alonso, et al. Beyond the imitation game: Quantifying and extrapolating the capabilities of language models. arXiv preprint arXiv:2206.04615, 2022.
> >
> > [2] Takeshi Kojima, Shixiang Shane Gu, Machel Reid, Yutaka Matsuo, and Yusuke Iwasawa. Large language models are zero-shot reasoners. arXiv preprint arXiv:2205.11916, 2022.

---

> > > ### Comment · Reviewer_y9rU · 2022-12-12
> > > **Thanks for the reminder and the additional work**
> > >
> > > Thanks for having specified the exact scope of the term "prompt-engineer" in your work.
> > >
> > > Within this scope, the title is more than legit. Still, I remain unsatisfied with this limited scope, which is not capturing what I consider the most interesting problems. Such as the "horse riding an astronaut" example.
> > > In the current context, as a reader, I would focus my attention on tasks where there is a large difference between human and APE performance. It would be a great achievement if human performance on these tasks was at random chance, while APE performs significantly better.
> > >
> > > This seems to be the case for some Big-Bench tasks, such as winowhy: Human prompt -12 while APE 12 (Random chance is 0 as far as I understand).
> > >
> > > I read in the paper (sec 4.2) that the human prompts correspond to the default human prompts in Big-Bench, so it seems from this -12 that the default performance on winowhy measured in the original Big-Bench paper should be below random chance. But this is not the case: most models are beyond random-chance on winowhy: https://github.com/google/BIG-bench/tree/main/bigbench/benchmark_tasks/winowhy
> > >
> > > I am probably missing something, but I was not able to figure it out.
> > >
> > > The authors kindly asked me to raise my rating. Given the above concerns, I do not feel confident in raising my 6. Fortunately, there seems to be an agreement to accept this paper, and increasing my rating is not vital.
> > >
> > > Thank you again and sorry for the late response.

---

### Author Response · Authors · 2022-11-18
**General Comments to All Reviewers (1/2)**

# Summary

We would like to thank all the reviewers for their time and effort in the review process. We appreciate that you found our work “interesting” (y9rU, 7jaz), “novel” (y9rU, 7jaz), “well written” (y9rU, 3MTv), and our experiments “well-done” (3MTv) with “significant improvements over the baseline” (3MTv). We’ve responded to each reviewer individually, uploaded a revised draft, and collected the below responses to general concerns. If you find our answers responsive to your concerns, we would be grateful if you would consider increasing your score.

> Several reviewers asked for evaluation on different, more challenging tasks.

In order to address any concerns with the empirical results in our paper, we expanded our evaluation to include two more benchmark suites (a total of 21+12 new tasks). We also re-ran our original instruction induction experiment with updated hyperparameters and achieve super-human performance on 24 out of 24 tasks.

## Big Bench Instruction Induction (BBII)

To address the common concern that our tasks may not be challenging enough, we first evaluate APE on a subset of BigBench [1]. We propose and curate BigBench Instruction Induction (BBII): a clean, and tractable subset of 21 tasks that satisfy the instruction induction format. Similar to the original Instruction Induction dataset, we select tasks that include both a human instruction (which we replace with our APE instruction), and a set of input/output pairs. The selected tasks cover many facets of language understanding and include all nine such problems from the BigBench-Hard Subset [2]. In particular, it includes emotional understanding, context-free question answering, reading comprehension, summarization, algorithms, and various reasoning tasks (e.g., arithmetic, commonsense, symbolic, and other logical reasoning tasks). We provide a detailed description of the task and our selection criteria in Appendix B.1.

We use APE to generate new prompts for the tasks in BIG-Bench Instruction Induction (BBII) and find that APE retains its strong performance on this new benchmark suite. When compared to human prompts, APE generated prompts improve or match zero-shot performance on 17 out of 21 tasks. We report the normalized preferred metric defined in [3]. Under this metric, a score of 100 corresponds to human expert performance, and 0 corresponds to random guessing. Note that a model can achieve a score less than 0 if it performs worse than random guessing on a multiple-choice task.

| Task                              | Human    | APE      |
|-----------------------------------|:----------:|:----------:|
| causal judgment                   | 18.0          | 18.0     |
| disambiguation qa                 | -0.4          | **5.6**  |
| dyck languages                    | 3.0           | **18.0** |
| epistemic reasoning               | 36.0          | **38.0** |
| gender inclusive sentences german | 13.0          | **22.0** |
| implicatures                      | 60.0          | 60.0     |
| linguistics puzzles               | 0.0           | 0.0      |
| logical fallacy detection         | **24.0**      | 12.0     |
| movie recommendation              | -2.7          | **12.0** |
| navigate                          | -8.0          | **12.0** |
| object counting                   | 2.0           | **44.0** |
| operators                         | **48.0**      | 47.0     |
| presuppositions as nli            | **13.0**      | 5.5      |
| question selection                | -2.6          | **-0.9** |
| ruin names                        | **1.3**       | -14.7    |
| snarks                            | 2.0           | **4.0**  |
| sports understanding              | 36.0          | 36.0     |
| tense                             | 84.0          | **85.0** |
| winowhy                           | -12.0         | **12.0** |
| word sorting                      | 11.0          | **30.0** |
| word unscrambling                 | 10.0          | **15.0** |

## Zero-Shot Chain-of-Thought (Optimizing “Let’s think step by step”)

Moreover, we follow [4] to evaluate APE on 12 reasoning tasks from four categories of reasoning tasks: arithmetic, commonsense, symbolic, and other logical reasoning tasks. As shown in the table below. APE is able to find a prompt “Let’s work this out in a step by step way to be sure we have the right answer” that ranks first among the hand crafted prompts in Table 4 of [4], where the previous SOTA (which remained the best known CoT-triggering prompt for ~6 months), “Let’s think step by step”, is the most famous recent example of prompt engineering. The APE-generated prompt improves text-davinci-002’s Zero-Shot-CoT performance on MultiArith from 78.7 to 82.0 and improves GSM8K from 40.7 to 43.0.

---

> ### Author Response · Authors · 2022-11-18
> **General Comments to All Reviewers (2/2)**
>
> ## Updated Instruction Induction Results
>
> In running further experiments with APE, we found significantly better hyperparameters (higher number of candidate instructions generated using several different sets of examples). We re-evaluated APE with these better hyperparameters on the 5 instruction induction tasks where APE underperformed the human baseline in the original submission. **We now find that APE can achieve human level performance on all of our original tasks.** APE achieves a solidly super-human performance on the instruction induction task with an IQM of 0.810 vs humans’ 0.749.
>
> |      Task Name      | APE Mean Acc (old) | APE Mean Acc (new) | APE - Human |
> |:-------------------:|:--------------:|:-------------:|:---------:|
> |    Second Letter    |      0.596     |      0.8      |   0.034   |
> |    Pluralization    |      0.984     |     0.996     |   -0.004  |
> |    Passivization    |      0.622     |      1.0      |   0.001   |
> | Sentence Similarity |      0.186     |     0.256     |   -0.01   |
> |      Membership     |      0.126     |     0.612     |   -0.001  |
>
>
> ## Changes made to the paper since initial submission
>
> Here is a comprehensive list of what has changed since our original submission
> - [New Experiment] We add an experiment on **21 Big-Bench** (See our selection criteria in Appendix B.1) instruction induction tasks to demonstrate the effectiveness of APE on more challenging tasks. (Paper Section 4.2) @y9rU, @3MTv, @7jaz
> - [New Experiment] We evaluate APE on **12 zero-shot CoT reasoning tasks** to demonstrate APE’s ability to optimize non-instruction prompts, demonstrating that it can be a valuable tool for many different types of prompting setups. (Paper Section 4.3) @y9rU, @3MTv, @7jaz
> - [Updated Experiment] We re-evaluate APE with better hyperparameters on the 5 instruction induction tasks where APE underperforms the human baseline in the initial submission. We find that APE can achieve human level performance on all those tasks. APE now achieves a **super-human performance on instruction induction tasks** with an IQM of 0.810 vs humans’ 0.749. @3MTv
> - [New Analysis] We add a cost analysis to Appendix D showing that APE can reduce the cost by condensing the few-shot examples to more concise instructions, saving costs by reducing tokens in the long-run. @y9rU
> - [New Analysis] We add three more analyses to investigate the role of large language models in generating instruction proposals, the role of meta prompts in model performance and the transferability of the generated instructions to different models. (Appendix C.4)
> - [Clarity] Update Figure 1 to include concrete examples. Update the format of prompt templates in the method section to improve readability (Figure 2, 3). Add all raw templates used for model prompting to Appendix Table 5. @7jaz
> - [Clarity] Clarify the motivation of reverse mode generation and compare it with forward mode template. @7jaz
> - [Clarity] Clarify the definition of “prompt engineer” to make it clear that APE only optimizes single prompts, not multiple prompts chained together in a pipeline. We also include discussion to make it clear that APE can be used as a tool by human prompt engineers to optimize individual parts of such pipelines. @y9rU
>
>
>
> [1] BIG-bench collaboration. Beyond the imitation game: Measuring and extrapolating the capabilities of language models. In preparation, 2021. URL https://github.com/google/BIG-bench/.
>
> [2] Mirac Suzgun, Nathan Scales, Nathaneal Scharli, Sebastian Gehrmann, Yi Tay, Hyung Won Chung, Aakanksha Chowdhery, Quoc V. Le, Ed H. Chi, Denny ZHou, and Jason Wei. Challenging BIG-Bench tasks and whether chain-of-thought can solve them. arXiv preprint arXiv:2210.09261, 2022. URL https://arxiv.org/abs/2210.09261.
>
> [3] Aarohi Srivastava, Abhinav Rastogi, Abhishek Rao, Abu Awal Md Shoeb, Abubakar Abid, Adam Fisch, Adam R Brown, Adam Santoro, Aditya Gupta, Adrià Garriga-Alonso, et al. Beyond the imitation game: Quantifying and extrapolating the capabilities of language models. arXiv preprint arXiv:2206.04615, 2022. URL https://arxiv.org/abs/2206.04615.
>
> [4] Takeshi Kojima, Shixiang Shane Gu, Machel Reid, Yutaka Matsuo, and Yusuke Iwasawa. Large language models are zero-shot reasoners. arXiv preprint arXiv:2205.11916, 2022. URL https://arxiv.org/abs/2205.11916.

---

### Public Comment · ~Saeed_Najafi1 · 2023-07-17
**Lack of Comparison with Discrete Instruction (prompt) Optimization Techniques**

Prompting an LLM to generate candidate instructions is an interesting idea, however the recent discrete prompt optimization techniques, starting from the initially generated instructions, can surpass human designed instructions or even this APE technique.

The experiments lack comparison with important baselines such as AutoPrompt Gradient-based search, RLPrompt, GrIPS for gradient-free editing of the instruction, etc.

None of these important baselines have been compared with this black-box search-rank method.

Strong Baselines from the NLP community:
- https://arxiv.org/abs/2205.12548
- https://arxiv.org/abs/2010.15980
- https://arxiv.org/abs/2203.07281

---

> ### Public Comment · ~Chris_K1 · 2023-10-07
> **AutoPrompt baseline**
>
> I was going through the paper and realized the same question; I agree AutoPrompt should have been included as comparable baseline. Thanks for asking before.

---

### Decision · Program_Chairs · 2023-01-20

**Decision:**

Accept: poster

**Justification For Why Not Higher Score:**

- The topic to better leverage LLMs is very popular and there are many related works in the area. Even in automatic prompt generation, prefix tuning and the work done by Honovich et al. (2022, Instruction induction: From few examples to natural language task descriptions.) can be viewed as pioneer in the direction, which slightly affects the novelty and impact of this work.


**Justification For Why Not Lower Score:**

- See the strength of the paper for more details. The paper is working on a trendy topic to better leverage LLMs and the topic is of wide spread interests in the commnuity. The authors also show promising results to generate prompt and demonstration automatically. Many researchers in the area can benefit from the works.

**Metareview: Summary, Strengths And Weaknesses:**

The paper proposed using large language models (LLM) to generate instruction and prompts. The prompts and instruction can then be used for in-context learning and leveraging other LLM to solve NLP tasks. In the proposed approach, a pool of instructions are first sampled by a LLM. These instructions are scored and selected as demonstration for other LLM to conduct in-context learning and solve tasks at inference time. The idea is novel and interesting. The authors also showed the effectiveness of proposed approach empirically. They compare the method in zero and few shot settings with 1) human-engineered prompts 2) LLM generated outputs without the scoring and selection process. Results on two benchmarks show that the proposed approach produces instructions that are as good as instructions written by humans and achieve performance comparable to 1) / better than 2). Comprehensive ablation and analysis were also conducted for readers to understand the importance of each component in proposed approach.

Strength of the paper:
1. The paper is clear and well written. It is easy to understand the problem, the method, and the experiments. It provides a lot of insights that can help other researchers who are working on similar area.
2. The proposed idea is very interesting and novel. Better leveraging LLM is a very popular and trendy area, and exploring methods to obtain better prompts is one of the core topic with obvious importance. This paper is denitely of interest to the community.
3. The method is sound. Experiments and ablations are well-designed. Results also show the effectiveness of proposed approach. Automatic prompt generation is relevant to prefix tuning, but offers more interpretable inference process. Performance comparable to human engineered prompt/demonstration also show the direction is promising.

Weakness of the paper:
1. Several reviewers suspect if the proposed approach can achieve what the title and abstract claimed that LLM can generate prompt automatically with quality comparable to human engineering. Specifically, although experiments indicated that the proposed approach yields promising results in tasks requiring surface-level understanding of language syntax, the proposed approach consistently performs worse than humans in harder tasks such as sentence similarity and membership. The authors did a good job in providing additional discussion and experiment results. The additional results suggest proposed approach is generalizable to harder tasks. Performance in tasks where APE yields suboptimal performance previously is also improved due to better hyperparameter tuning. The additional explanation on the choice of titles, more experiment results supporting the efficacy of proposed approaches, and value propositions of APE compared to few-shot learning significantly mititgate reviewers' concerns.
2. Reviewers also raised some questions about the assumptions/intuitions adopted in the method design as well as the selection of benchmarks. The authors answered those questions properly in the discussion phase.

**Note From Pc:**

if the above contains the word "oral" or "spotlight" please see: "oral" presentation means -> notable-top-5% and "spotlight" means -> notable-top-25%. As stated in our emails, we are disassociating presentation type from AC recommendations